# Microbiota mediated plasticity promotes thermal adaptation in the sea anemone *Nematostella vectensis*

Laura Baldassarre [1,2], Hua Ying[3], Adam M. Reitzel [4], Sören Franzenburg [5] & Sebastian Fraune [1✉]

At the current rate of climate change, it is unlikely that multicellular organisms will be able to adapt to changing environmental conditions through genetic recombination and natural selection alone. Thus, it is critical to understand alternative mechanisms that allow organisms to cope with rapid environmental changes. Here, we use the sea anemone *Nematostella vectensis*, which has evolved the capability of surviving in a wide range of temperatures and salinities, as a model to investigate the microbiota as a source of rapid adaptation. We long-term acclimate polyps of *Nematostella* to low, medium, and high temperatures, to test the impact of microbiota-mediated plasticity on animal acclimation. Using the same animal clonal line, propagated from a single polyp, allows us to eliminate the effects of the host genotype. The higher thermal tolerance of animals acclimated to high temperature can be transferred to non-acclimated animals through microbiota transplantation. The offspring fitness is highest from F0 females acclimated to high temperature and specific members of the acclimated microbiota are transmitted to the next generation. These results indicate that microbiota plasticity can contribute to animal thermal acclimation and its transmission to the next generation may represent a rapid mechanism for thermal adaptation.

[1] Institute for Zoology und Organismic Interactions, Heinrich-Heine Universität Düsseldorf, Düsseldorf, Germany. [2] Istituto Nazionale di Oceanografia e di Geofisica Sperimentale—OGS, Sezione di Oceanografia, Trieste, Italy. [3] ANU Research School of Biology, The Australian National University, Canberra, ACT, Australia. [4] Biological Sciences, University of North Carolina at Charlotte, Charlotte, NC, USA. [5] Institute for Clinical Molecular Biology, Kiel University, Kiel, Germany. ✉email: fraune@hhu.de

Changes in the climate are proceeding worldwide at a rate never registered before and temperatures will rise dramatically in the coming decades. Species able to migrate could move toward new favorable areas, but those that have limited dispersal capacities or are sessile will have only two options: adaptation or extinction. Since the Modern Synthesis[1], the balance between mutation and natural selection has been considered the main source of phenotypic novelty and thus of the ability of populations to adapt to new environmental conditions. However, in some organisms acclimation to environmental change can occur within one generation.

Changes in gene expression have long been considered the most important explanation for this ability[2,3], although genetics alone cannot fully explain an organism's phenotype. Unlike the genes and regulatory regions of the genome, the microbial composition can be rapidly modified by environmental cues, and may thus represent a mechanism for rapid acclimation and adaptation of individuals to a changing environment[4–7]. Recently, the microbiota-mediated transgenerational acclimatization concept was proposed[8], suggesting that changes in microbiota assemblages may be passed on through generations to confer long-lasting fitness advantages to changing environments by individuals and populations.

To be able to separate host genetic contribution from the microbial contributions to thermal acclimation, we here resort to the model system *Nematostella vectensis*[9]. *N. vectensis*, an anthozoan cnidarian, is a sedentary predator that resides exclusively in estuaries and brackish water environments, where it lives burrowed in sediments[10]. It is a widespread species that has been found in both the Pacific and Atlantic coasts of the US and of the UK. In their natural habitats, wild populations of *N. vectensis* experience high variations of salinity, temperature and pollutants[11–16]. Under laboratory conditions, all the developmental stages are procurable weekly and spawning is induced by a shift in temperature and exposure to light[17]. *N. vectensis* can be easily cultured in high numbers[13] and clonally propagated to eliminate genetic confounding effects. A detailed analysis of its microbiota revealed that *N. vectensis* harbors a specific microbiota whose composition changes in response to different environmental conditions and among geographic locations[18]. Recently, it was shown that female and male polyps transmit different bacterial species to the offspring and that additional symbionts are acquired from the environment during development[19]. Furthermore, a protocol based on antibiotic treatment was established to generate germ-free animals that allow controlled recolonization experiments to be conducted[20]. Altogether, these characteristics make the sea anemone *N. vectensis* a uniquely informative marine model organism to investigate the effects of bacterial plasticity on thermal acclimation[5].

Here, we use a clonal lineage of *N. vectensis* to characterize the physiological and microbial plasticity of the holobiont under different long-term thermal acclimation regimes, while eliminating the variability due to different host genotypes. Using microbial transplantations to non-acclimated polyps, we prove the ability of acclimated microbes to confer resistance to thermal stress. We further show that polyps acclimated to high temperatures pass on higher fitness to the next generation.

Altogether, we provide strong evidence that microbiota-mediated plasticity contributes to the adaptability of *N. vectensis* to high temperature and that the transmission of acclimated microbiota represents a mechanism for rapid adaptation.

## Results

**Long-term acclimation at high temperature increases heat resistance in *Nematostella vectensis*.** Before starting the acclimation experiment, we propagated a single female polyp to 150 clones and split these clones into 15 different cultures with 10 clonal animals each, to ensure the same genotype in all acclimation regimes. We further propagated these animals to 50 animals per culture and constantly maintained this number throughout the experiment. Subsequently, we acclimated these independent cultures at low (15 °C), medium (20 °C), and high temperature (25 °C) (five cultures each) for 3 years (161 weeks) (Fig. 1).

After 40 weeks of acclimation (woa) we tested, for the first time, the heat tolerance of acclimated polyps as a proxy for acclimation. We individually incubated polyps of each acclimated culture in ten replicates for 6 h at 40 °C and recorded their mortality (Fig. 2a). Already after 40 woa, significant differences in the mortality rates of clonal animals were detectable. While all animals acclimated to low temperature died after the heat stress, animals acclimated at 20 °C showed a significant higher survival rate of 70%. Animals acclimated at 25 °C showed a survival rate of 30%, although this was not significantly different from the survival rate at 15 °C (Fig. 2a). We repeated the measurement of heat tolerance 2 years later (132 woa). Interestingly, we observed a drastic increase in fitness in animals acclimated at high temperature, while the animals acclimated at 15 °C and 20 °C showed 100% mortality (Fig. 2a).

We also monitored the mortality rate in the acclimated cultures throughout 161 weeks (Fig. 2b). While the mortality in cultures acclimated at 15 °C and 20 °C was below 0.5 polyps per week, the mortality rate was significantly reduced in cultures acclimated at 25 °C. An additional phenotypic difference between the acclimated animals was the clonal growth, as animals acclimated at 25 °C propagated asexually nearly seven times more than animals acclimated at 15 °C (Fig. 2c). This may explain the differences in body size, where animals acclimated at 15 °C were more than three times bigger than the animals acclimated at 20 and 25 °C (Fig. 2d). The different ATs also affected the fecundity of the animals: the polyps acclimated at the high ATs showed a significantly higher number of spontaneous spawning events recorded along the whole course of the experiment, compared with the 15 °C acclimated animals that never spawned if not artificially induced (Supplementary Fig. 1).

These results indicate that *N. vectensis* possesses remarkable plasticity at long-term temperature acclimation realized through differences in thermal tolerance, body size, asexual propagation, and fecundity. In the following, we analyzed the associated microbiota and host transcriptomic responses as a source of thermal acclimation in *N. vectensis*.

**Thermal acclimation leads to dynamic, but consistent changes in the microbiota.** To monitor the dynamic changes in the associated microbiota of acclimated animals, we sampled single polyps from each of the 15 clonal cultures at 40, 84 and 132 woa and compared their associated microbiota by 16 S rRNA sequencing (Fig. 1). To determine the impact of AT and sampling time point on the assemblage of the bacterial community, we performed principal coordinates analysis (PCoA) (Fig. 3a, b).

While principal component 1 (PC1) mostly separates samples according to the AT (Fig. 3a), PC2 correlates with the different sampling time points (Fig. 3b). Using five different β-diversity metrics, we found that bacterial colonization is significantly influenced by both AT and woa (Table 1).

Assigning the different microbial communities by sampling time points revealed a shared clustering after 84 and 132 woa (Fig. 3b), suggesting a stabilization within the microbial communities after around 2 years of acclimation. In contrast, assigning the samples by AT revealed a clear clustering of the microbial communities (Fig. 3a) with the bacterial communities

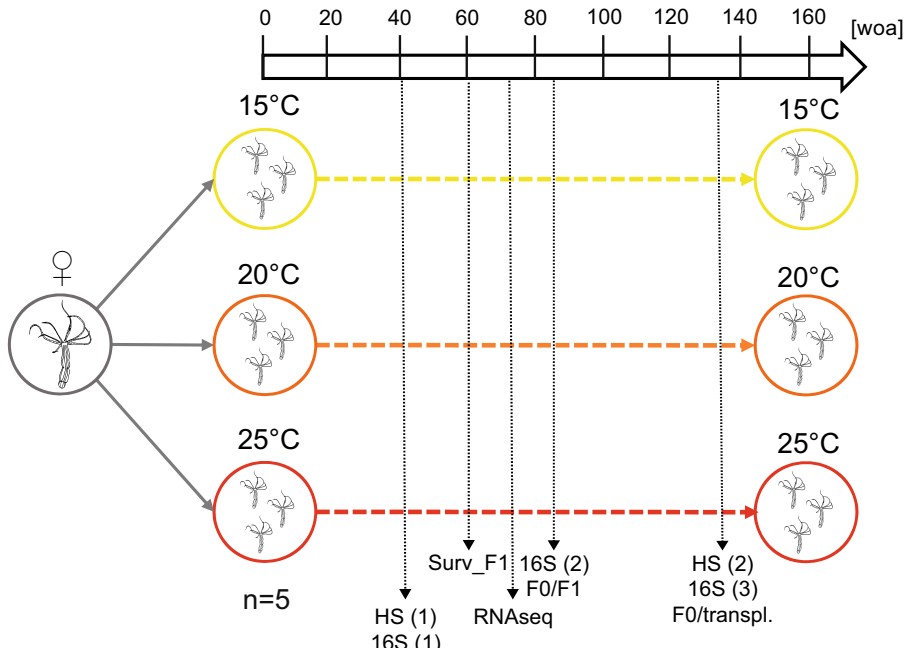

**Fig. 1 Experimental setup.** A single female polyp from the standard culture conditions (16‰, 20 °C) was isolated and propagated via clonal reproduction. When a total of 150 clones was reached, they were split into 15 different culture boxes of 10 animals each. The boxes were put at three different acclimation temperatures (ATs) (15, 20, and 25 °C, $n = 5$) and the number of animals/box was kept equal to 50. Heat stress experiments (HS) (6 h, 40 °C) were performed at 40 ($n = 10$) and 132 ($n = 5$) weeks of acclimation (woa). Sexual reproduction was induced at 60 and 84 woa for the juvenile survival test (Surv_F1, $n = 10$) and the vertical transmission experiment (F0/F1, $n = 5$). At 40, 84 and 132 woa samples were collected for 16S sequencing (16S, $n = 5$); at 76 woa sampling for RNA sequencing was performed ($n = 5$).

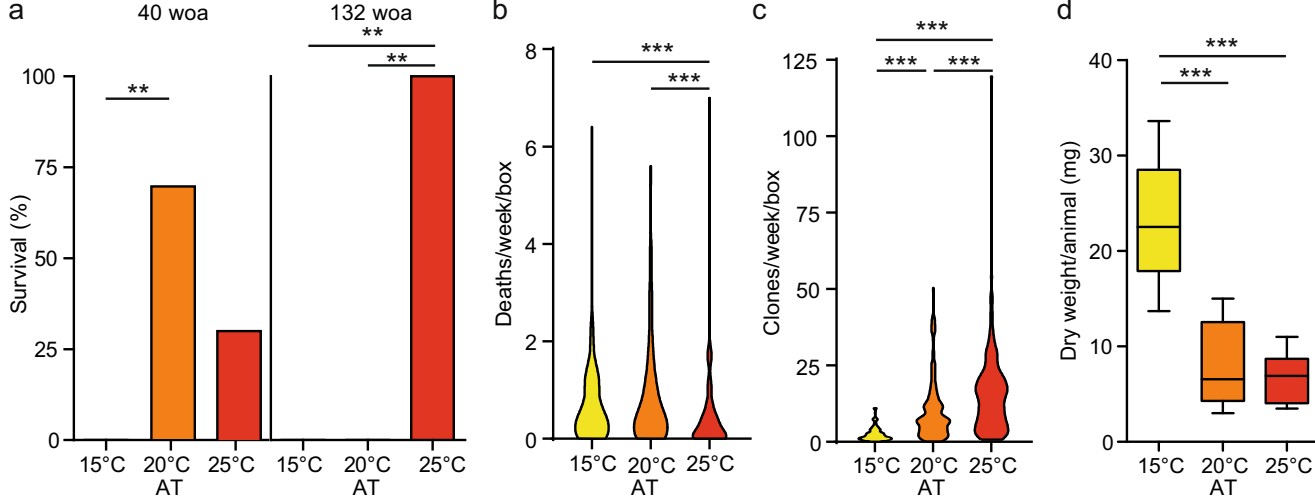

**Fig. 2 Phenotypic plasticity in response to thermal acclimation. a** Survival of acclimated polyps after heat stress (40 °C, 6 h). Statistical analyses were performed by a Fisher's exact test ($n = 10$ (40 woa), $**p = 0.0031$), $n = 5$ (132 woa, $**p = 0.0079$). **b** Average of dead polyps per week per box throughout the whole acclimation period ($n = 5$, 161 woa) (Kruskal–Wallis test followed by Dunn's post hoc comparisons, $H = 24.09$, $***p < 0.001$). **c** Average of clones generated per week per 50 animals throughout the whole acclimation period, $n = 5$, Kruskal–Wallis test followed by Dunn's post hoc comparisons, $H = 191.6$, $***p < 0.001$. **d** Dry weights of acclimated polyps at 161 woa, $n = 10$, Kruskal–Wallis test followed by Dunn's post hoc comparisons, $H = 19.01$, $***p < 0.001$. Box plots are presenting center line, median; box limits, upper and lower quartiles; whiskers, 1.5x interquartile range.

acclimated at 20 °C clustering between the two extremes (15 °C and 25 °C). This indicates that the three different ATs induced differentiation of three distinct microbial communities since the beginning of the acclimation process and that this differentiation is more pronounced between the extreme ATs. While most bacterial groups maintain a stable association with *N. vectensis* (Fig. 3c), bacteria that contribute to the differentiation at the end of the acclimation process are Alphaproteobacteria, which

significantly increase at high temperatures (one-way ANOVA, $F = 17.27$, $p = 0.0004$) (Supplementary Table 1), and Epsilonproteobacteria, which significantly increase at low temperature (one-way ANOVA, $F = 25.96$, $p < 0.0001$) (Fig. 3c, Supplementary Table 2).

Using the Binary-Pearson distance matrix, we calculated the distances between samples within all three acclimation regimes (Fig. 3d) and sampling time points (Fig. 3e). Continuous

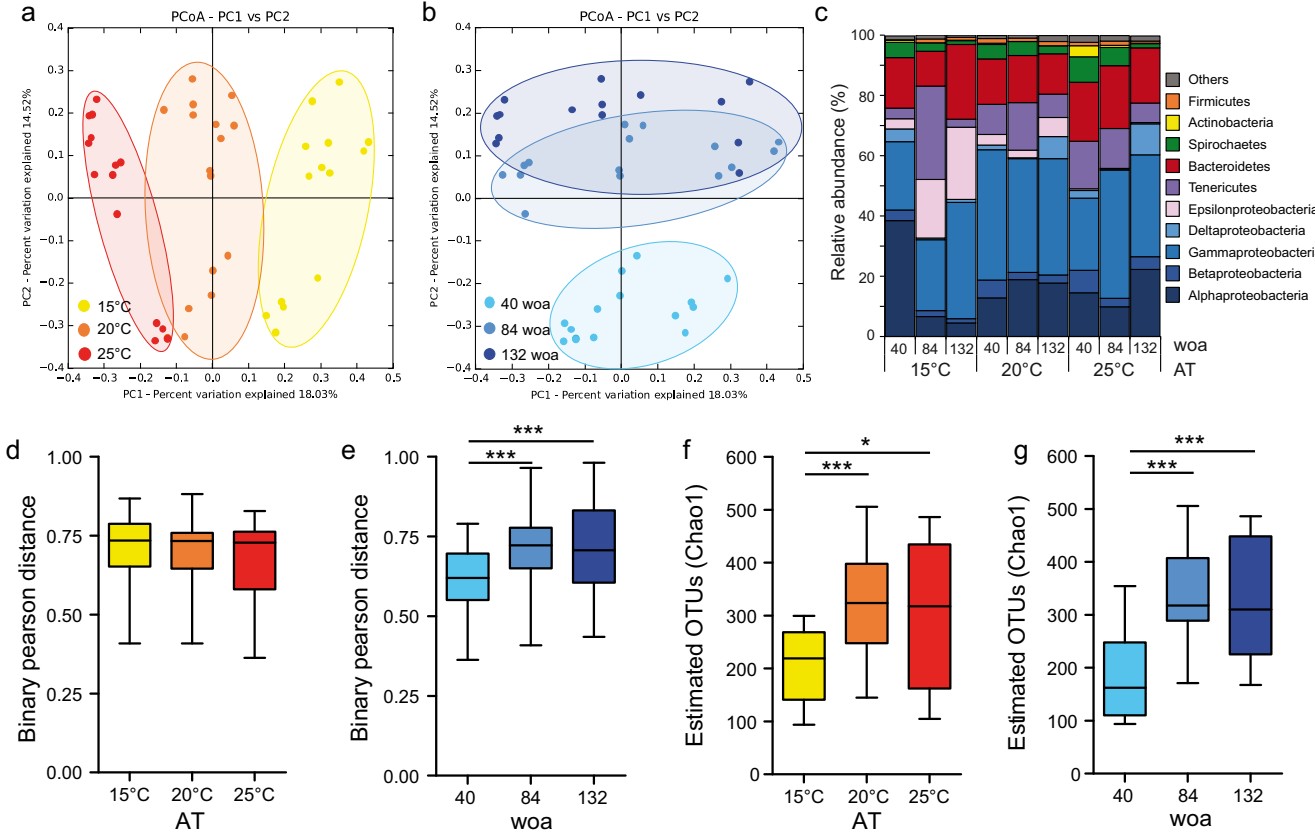

**Fig. 3 Bacterial community changes in response to thermal acclimation. a** PCoA (based on the binary-Pearson metric, sampling depth = 3600) illustrating similarity of bacterial communities based on acclimation temperature (AT). **b** PCoA (based on Binary-Pearson metric, sampling depth = 3600) illustrating similarity of bacterial communities based on weeks of acclimation (woa). Note: in **a** and **b** the same data are plotted, but with different color codes. **c** Relative abundances of principal bacterial groups, the abundances were summarized under the relative higher taxonomic categories (classes and phyla) and reported as percentages of the total. **d**, **e** β-diversity distances within each AT (**d**) and within woa (**e**), statistical analyses were performed using a non-parametric Kruskal–Wallis test followed by Dunn´s post hoc comparisons **e** $H = 43.66$; ***$p < 0.001$. **f** α-diversity (Chao1) comparison by AT (max rarefaction depth = 3600), statistical analyses were performed using a non-parametric Kruskal–Wallis test followed by Dunn's post hoc comparisons ($H = 9.801$; *$p = 0.011$, ***$p < 0.001$). **g** α-diversity (Chao1) comparison by woa (max rarefaction depth = 3600), statistical analyses were performed by using one-way ANOVA followed by Tukey's post hoc comparisons ($F = 12.036$; ***$p < 0.001$). Box plots are presenting center line, median; box limits, upper and lower quartiles; whiskers, 1.5x interquartile range.

**Table 1 Statistical analysis determining the influence of AT and woa on bacterial colonization.**

| Parameter | beta-diversity metric | Adonis | | Anosim | |
|---|---|---|---|---|---|
| | | $R^2$ | *p*-value | *R* | *p*-value |
| AT | Binary-Pearson | 0.208 | 0.001 | 0.544 | 0.001 |
| | Bray–Curtis | 0.219 | 0.001 | 0.466 | 0.001 |
| | Pearson | 0.256 | 0.001 | 0.360 | 0.001 |
| | Weighted-Unifrac | 0.147 | 0.001 | 0.238 | 0.001 |
| | Unweighted-Unifrac | 0.193 | 0.001 | 0.521 | 0.001 |
| woa | Binary-Pearson | 0.230 | 0.001 | 0.608 | 0.001 |
| | Bray–Curtis | 0.199 | 0.001 | 0.372 | 0.001 |
| | Pearson | 0.217 | 0.001 | 0.277 | 0.001 |
| | Weighted-Unifrac | 0.149 | 0.001 | 0.173 | 0.001 |
| | Unweighted-Unifrac | 0.192 | 0.001 | 0.498 | 0.001 |
| Number of permutations = 999. | | | | | |

acclimation under the different temperature regimes revealed no differences in the within-treatment distances (Fig. 3d), indicating similar microbial plasticity at all three ATs. In contrast, Binary-Pearson distances of the different sampling time points significantly increased between 40 and 84 woa (Fig. 3e) and stabilized between 84 and 132 woa. The α-diversities of bacteria associated with acclimated polyps were significantly higher at 20 and 25 °C, compared to those associated with polyps acclimated at 15 °C (Fig. 3f). As for the β-diversity, the α-diversity was significantly increasing within the first 84 woa and stabilized between 84 and 132 woa (Fig. 3g).

Altogether, these results show that the microbiota of *N. vectensis* reacts plastically to environmental changes. The microbial composition changes stabilize within two years of acclimation indicating a new homeostatic bacterial colonization status.

**Thermal acclimation induces a robust tuning of host transcriptomic profiles.** To evaluate the contribution of host transcriptional changes to the observed increased thermal tolerance in animals acclimated at high temperature, we analyzed gene expression profiles of *N. vectensis* after 75 woa (Fig. 1). We sampled from each replicate culture one animal, extracted its mRNA and sequenced it by Illumina HiSeq 4000. The constant acclimation at 15, 20, and 25 °C induced a robust tuning of the host transcriptomic profiles (Fig. 4a).

From pairwise comparisons, we determined the differentially expressed (DE) genes (Fig. 4b) in all acclimated animals. While

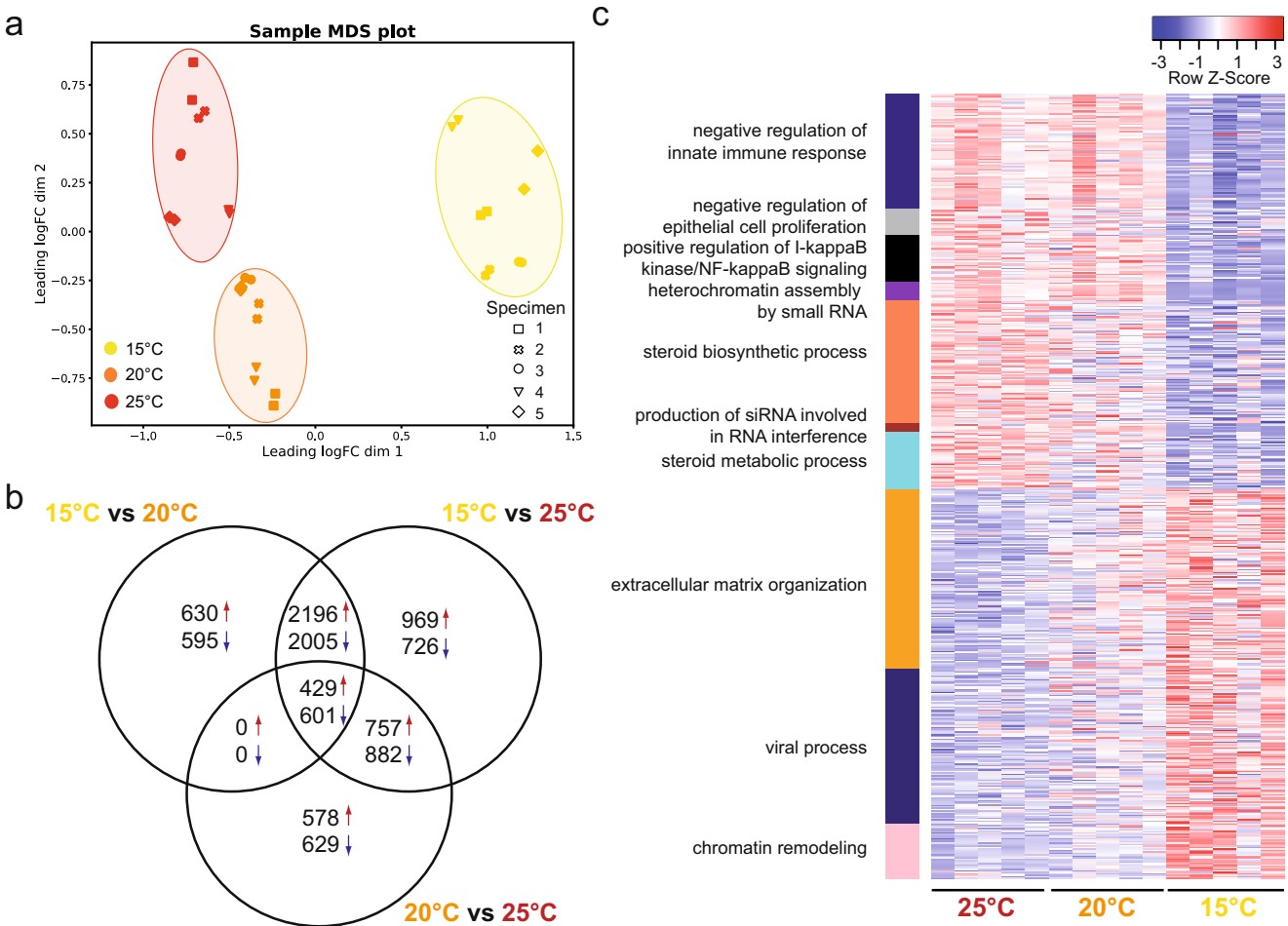

**Fig. 4 Host transcriptome changes after thermal acclimation. a** MDS plot showing clustering of the transcriptome profiles according to the AT (samples were sequenced in technical replicates, indicated by the different symbols). **b** Venn diagram showing the numbers of differentially expressed genes from pairwise comparisons between the three ATs. **c** Heat-map of differentially expressed genes in significantly enriched GO term categories from the comparison between 15 and 25 °C acclimated polyps.

the comparison of transcriptomic profiles from polyps acclimated at 15 and 25 °C revealed the highest number of DE genes, the comparison of 20 and 25 °C acclimated animals revealed the lowest number of DE genes. In all three comparisons, we observed a similar fraction of up- and down-regulated DE genes (Fig. 4b).

To retrieve molecular processes and signaling pathways enriched at the different ATs, we performed a gene ontology (GO) enrichment analysis and concentrated on GO categories significantly enriched in the comparison between 15 and 25 °C acclimated polyps (Fig. 4c and Supplementary Data 1). Animals acclimated to high temperature significantly increased expression in genes involved in innate immunity, gene regulation, epithelial cells proliferation, steroid biosynthesis, and metabolism (Fig. 4c and Supplementary Data 1). While genes associated with enriched GO categories show opposite expression levels at 15 and 25 °C, an intermediate expression level was evident in the animals acclimated at 20 °C (Fig. 4c). The animals acclimated to low temperature showed upregulation of genes associated with viral processes, which seems to be compatible with their general lower viability.

**Transplantation of acclimated microbiota induces differences in heat tolerance.** To separate the effects of transcriptomic from bacterial adjustments on the thermal tolerance of acclimated

polyps, we performed microbial transplantation experiments. We generated axenic non-acclimated animals (recipients) and recolonized these animals with the microbiota of long-term acclimated polyps (donors) from the same clonal line. We maintained recipient animals for 1 month at 20 °C to allow the adjustment of stable colonization.

The 16S rRNA gene sequencing of donor and recipient animals and subsequent PCoA of the recipient polyps revealed that they grouped according to the AT of the donor microbiota 1 month after transplantation (Fig. 5a and Table 2). To evaluate the rate of vertical bacterial transmission we included the donor samples into the PCoA (Supplementary Fig. 2 and Supplementary Table 3). While principal component 1 explained the bacterial variation due to transplantation, principal component 3 explained the bacterial variation due to the difference in acclimation temperature in donor polyps (Supplementary Fig. 2a-c). In addition, presence-absence analysis based on the OTU read table revealed that recipient polyps received a high proportion of differentially abundant bacteria from the corresponding donor polyps during transplantation (Supplementary Data 2). Nevertheless, not all bacteria could be transplanted with the same efficiency. Recipient polyps showed a reduction of bacterial α-diversity by approximately 30% compared to the donor polyps (Supplementary Fig. 2-d). While most bacterial classes were present in similar proportions in donor and recipient polyps,

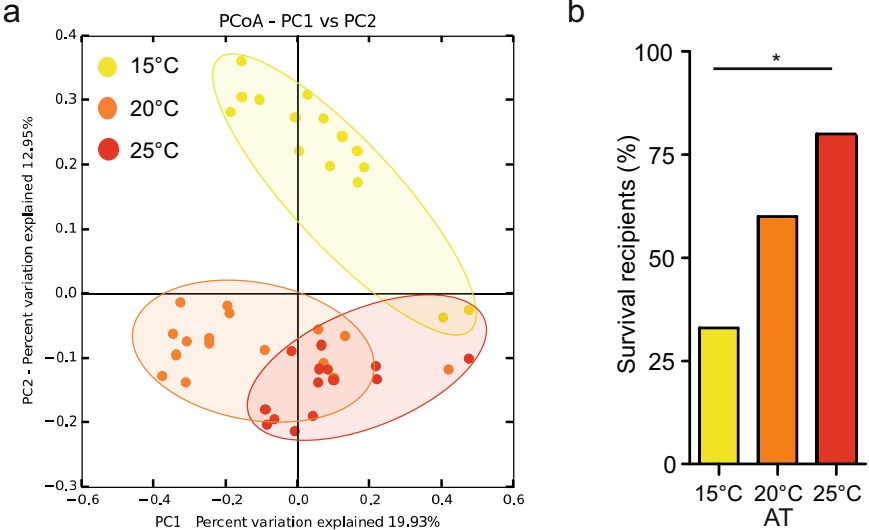

**Fig. 5 Transplantation of acclimated microbiota confers thermal resistance. a** PCoA (based on the binary-Pearson metric, sampling depth = 3600) illustrating similarity of recipient bacterial communities based on AT of donor microbiota. **b** Heat stress (40 °C, 6 h) survival of recipient polyps. Statistical analyses were performed by pairwise Fisher's exact test ($n = 15$, *$p = 0.025$). Box plots are presenting center line, median; box limits, upper and lower quartiles; whiskers, 1.5x interquartile range.

**Table 2 Statistical analysis determining the influence of donor polyps' AT on recipient microbiota.**

| Parameter | beta-diversity metric | Adonis | | Anosim | |
|---|---|---|---|---|---|
| | | $R^2$ | $p$-value | $R$ | $p$-value |
| Donors' AT | Binary-Pearson | 0.199 | 0.001 | 0.486 | 0.001 |
| | Bray–Curtis | 0.183 | 0.001 | 0.346 | 0.001 |
| | Pearson | 0.165 | 0.001 | 0.194 | 0.001 |
| | Weighted-Unifrac | 0.161 | 0.001 | 0.272 | 0.001 |
| | Unweighted-Unifrac | 0.184 | 0.001 | 0.416 | 0.001 |

Number of permutations = 999.

Epsilonproteobacteria did not appear to be transplantable (Supplementary Fig. 2-e).

Subsequently, we tested the recipient animals for their heat tolerance as previously performed for the acclimated animals. The recipient animals showed clear differences in mortality after heat stress depending on the microbial source used for transplantation. A significant gradient in survival was evident from the animals transplanted with the 15 °C acclimated microbiota (33%) to those transplanted with the 25 °C acclimated microbiota (80%) (Fig. 5b). The animals transplanted with the 20 °C acclimated microbiota showed an intermediate survival (60%).

These results indicate that the high thermal tolerance of animals acclimated to high temperature can be transferred to non-acclimated animals by microbiota transplantation alone. Therefore, we conclude that microbiota-mediated plasticity provides a rapid mechanism for a metaorganism to cope with environmental changes.

Through LEfSe analysis, we were able to detect bacterial OTUs differentially represented between the polyps acclimated at 15 and 25 °C, and in the corresponding transplanted animals (Supplementary Table 4). These bacteria belong to the families Phycisphaeraceae, Flavobacteriaceae, Emcibacteraceae, Rhodobacteraceae, Methylophilaceae, Francisellaceae, Oceanospirillaceae, and Vibrionaceae, which are known to include various commensals, symbionts, and pathogens of marine organisms.

Therefore, the OTUs overrepresented in the 25 °C microbiota may constitute good candidates for providing thermal resistance to their host.

**Acclimated microbiota and higher fitness are transmitted to the next generation**. In the next step, we tested if the higher temperature resistance of F0 animals acclimated to high temperature is transmitted to the offspring (F1 generation). Therefore, two female polyps from each long-term acclimated culture and one non-acclimated male polyp were induced separately for spawning. All oocyte-packs were fertilized with the sperm of the same male polyp, split into 3 parts, counted and let develop for 1 month at the 3 different developing temperatures (DTs) in a full factorial design (Fig. 6a).

After 1 month of development, the survived juvenile F1 polyps were counted and corresponding survival rates were calculated (Fig. 6b). The offspring from the polyps acclimated at 25 °C showed a significantly higher overall survival rate compared to the offspring from polyps acclimated at medium and low temperature. In contrast, the offspring of polyps acclimated at 15 °C showed the lowest survival rate at 25 °C DT (Fig. 6b). In a second step, the F0 and the juvenile F1 polyps were subjected to 16S rRNA sequencing to evaluate the transmission of acclimated microbes to the next generation. PCoA of F1 animals revealed a significant clustering according to both F1 DT and F0 AT (Fig. 6c, d and Table 3). While, on average, around 50% of bacterial variation can be explained by the DT of the juvenile polyps, around 20% of the bacterial colonization in juveniles can be explained by the acclimation temperature of the F0 polyps (Table 3). The comparison of F0 and F1 samples in PCoA revealed a clustering between F0 and F1 samples along PC1 (Supplementary Fig. 3-a), indicating that differential bacterial communities colonize juvenile and adult polyps, as described in earlier studies[18,19]. In contrast, PC2 separated samples according to AT in both the F0 and the F1 samples (Supplementary Fig. 3-a and Supplementary Table 5), indicating the successful transplantation of parts of the acclimated bacterial communities.

The successful transmission is also indicated by the concordance of bacterial groups present in the F0 and F1 animals (Supplementary Fig. 3-b). For the identification of individual

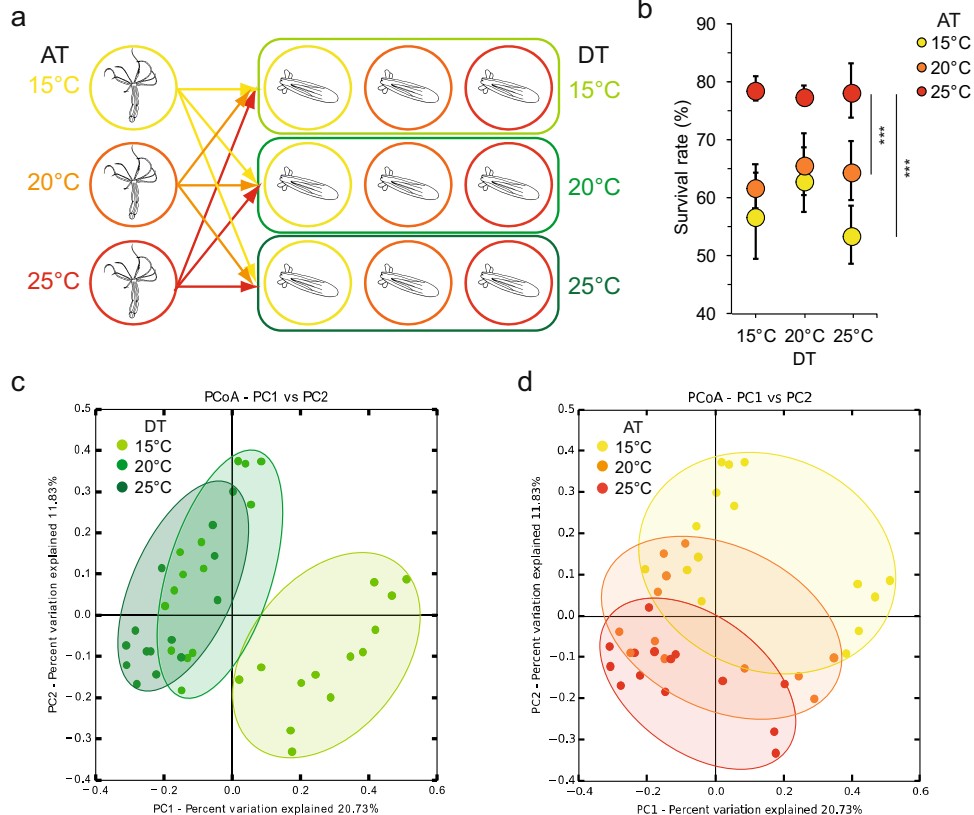

**Fig. 6 Transmission of thermal tolerance to the offspring. a** Experimental scheme: acclimated females from each AT were induced for sexual reproduction. All oocyte-packs were fertilized with the sperms from a single male polyp from the standard conditions. After fertilization, each embryo pack was split into 3 parts and placed at different DT (15, 20, or 25 °C). After 1 month of development, survival rate and bacterial colonization were analyzed. **b** Offspring survival rate (ratio between the initial number of oocytes and survived juvenile polyps were calculated), a Kruskal–Wallis test was performed followed by Dunn´s post hoc comparisons ($n = 10$; $H = 32.658$; ***$p < 0.001$), average values with error bars indicating SE are presented. **c** PCoA (based on Binary-Pearson metric, sampling depth = 24,500) illustrating similarity of bacterial communities according with F1 DT. **d** PCoA (based on binary-Pearson metric, sampling depth = 24,500) illustrating similarity of bacterial communities according with F0 AT. Note: in **c** and **d** the same data are plotted, but with different color code.

**Table 3 Statistical analysis determining the influence of F0 females' AT and F1 DT on bacterial colonization of F1 polyps.**

| Parameter | beta-diversity metric | Adonis | | Anosim | |
|---|---|---|---|---|---|
| | | $R^2$ | $p$-value | $R$ | $p$-value |
| F0 polyps' AT | Binary-Pearson | 0.170 | 0.001 | 0.373 | 0.001 |
| | Bray–Curtis | 0.133 | 0.001 | 0.237 | 0.001 |
| | Pearson | 0.139 | 0.001 | 0.166 | 0.002 |
| | Weighted-Unifrac | 0.093 | 0.024 | 0.091 | 0.010 |
| | Unweighted-Unifrac | 0.116 | 0.001 | 0.148 | 0.005 |
| DT | Binary-Pearson | 0.262 | 0.001 | 0.696 | 0.001 |
| | Bray–Curtis | 0.260 | 0.001 | 0.621 | 0.001 |
| | Pearson | 0.338 | 0.001 | 0.542 | 0.001 |
| | Weighted-Unifrac | 0.300 | 0.001 | 0.408 | 0.001 |
| | Unweighted-Unifrac | 0.211 | 0.001 | 0.413 | 0.001 |

Number of permutations = 999.

OTUs potentially transmitted from F0 to F1 animals, we performed a presence/absence analysis (Supplementary Data 3) revealing the differential transmission of bacterial species from F0 female polyps acclimated at different temperatures. Using LEfSe analysis, we found that bacterial OTUs were overrepresented in polyps acclimated at 15, 20, and 25 °C as well as in the corresponding F1 animals (Supplementary Table 6). While F0

female polyps acclimated at 15 °C mainly transmitted Bacteroidetes and Alphaproteobacteria, polyps acclimated at 25 °C transmitted mainly Gammaproteobacteria to their offspring (Supplementary Table 6).

These results demonstrate that acclimation at high temperature of the F0 generation improved the overall fitness of its offspring. The fact that also specific members of the acclimated microbiota are transmitted and persisted in the F1 generation suggests that vertically transmitted acclimated bacteria can be adaptive to high temperature.

## Discussion

**Long-term acclimation promotes heat tolerance in *N. vectensis*.** The ability of marine animals to adapt to future thermal scenarios is of pivotal importance for the maintenance of biodiversity and ecosystem functioning. Recent studies indicate that sessile marine animals, like corals, sponges or anemones, could adapt more rapidly to climate change than expected[3,21–25]. Recent and long-term observations in the field displayed higher heat tolerance of corals pre-exposed to thermal stress compared to unexposed ones and showed that wild populations are slowly becoming less sensitive than they were in the past[26–28]. In our study, the host's thermal resistance showed an increase with the acclimation time. It is important to point out that the standard culture temperature for *N. vectensis* in the lab is 20 °C. The animals maintained at 20 °C, therefore, have been acclimated to this condition for a long

time and this might explain their highest survival at 40 woa. Interestingly, the animals acclimated at 15 °C showed at both time points 100% mortality, indicating that these animals would not be able to survive extreme high temperature events.

Our results are consistent with other studies that investigated the acclimation capacity of corals in lab experiments. Pre-acclimated individuals of *Acropora pruinosa*, a scleractinian coral, showed lower signs of bleaching when exposed to successive heat stress, in comparison to the not-pre-acclimated ones[25]. Also in the field, *Acropora hyacinthus* showed less mortality after heat stress when acclimated to wide temperature fluctuations, than when acclimated to less variable environments[29]. These different resistances are correlated to adaptive plasticity in the expression of environmental stress response genes[30] and the presence of an advantageous microbiota[31], but a causative relation was not shown in both cases. Our study separates the contribution of the microbiota to temperature acclimation from host effects by performing microbial transplantation experiments on a single host genotype.

### Microbiota plasticity promotes metaorganism acclimation.
Shifts in the composition of bacterial communities associated with marine animals in response to changes in environmental factors (i.e., temperature, salinity, pH, light exposure, oxygen, $CO_2$ concentrations, etc.) have been demonstrated in numerous studies[18,32–38]. In some cases these changes in microbiota composition correlated with a higher fitness of acclimated animals[36], but causal connections are rare. An experimental replacement of a single bacterium and subsequent demonstration of acquired heat tolerance by the host was only shown in aphids[39].

To infer if and to what extent the acclimated microbiota confers thermal resistance, we performed transplantation experiments of microbiota from acclimated animals to non-acclimated ones. These experiments proved that polyps transplanted with the microbiota from animals acclimated at 25 °C for 132 woa acquired a higher thermal tolerance than those transplanted with the 15 °C acclimated microbiota. It is important to point out that the animals selected as recipients for this experiment were all clones of the same age, size and shared the same life history, since they came from the same culture box, and belonged to the same clonal line as the acclimated donors. With this experimental setup, we were able to disentangle microbiota contribution to thermal acclimation from host genotype effects and proved that acclimated bacteria can act as heat tolerance-promoting bacteria.

Microbial community acclimation is a highly dynamic process that began in the first few weeks after the environmental change, and most adjustments in bacterial diversity occurred by 84 woa. Afterward, the microbial community likely reached a stable and homeostatic state. Previous studies on corals[40,41] demonstrated the presence of a "core microbiota", defined as a group of microbial species that are either persistent over time and/or in different environments or locations and are less sensitive to changes in the surrounding environment. Members of the core microbiota may not necessarily represent the most abundant groups of the community but are hypothesized to exert pivotal functions for the maintenance of the holobiont homeostasis. In contrast, there is a "dynamic microbiota" that varies by species, habitat, and life stage and is likely a product of stochastic events or a response to changing environmental conditions[41]. Also in *N. vectensis*, it appears that during the acclimation process, a core microbiota remained stable in all acclimated polyps, while a more dynamic part of the microbiota changed, either increasing or decreasing the abundance of certain species. The increase in α-diversity indicates either the acquisition of new bacterial species from the environment or a higher evenness in species

abundances, where OTUs that were rare at the beginning of the experiment and at lower temperature, became more abundant and thus detectable. The acquisition of new bacterial species during lab experiments appears unlikely since the polyps are isolated from their natural environment. Nevertheless, the acclimated animals are not maintained under sterile conditions and thus an exchange of microbial species with the culture medium and from the food supply cannot be excluded. As already pointed out in numerous studies[42–44], higher microbial diversity enhances the ability of the host to respond to environmental stress by providing additional genetic diversity.

In addition to the changes in species composition and relative abundances, the associated microbial species can evolve much more rapidly than their multicellular host[8]. Rapidly dividing microbes are predicted to undergo adaptive evolution within weeks to months[45]. Therefore, an adaptation of the host can also occur via symbiont acquisition of novel genes[46], via mutation and/or horizontal gene transfer[8]. Even if the abundance of a certain bacterial species did not change significantly between the different ATs, it is possible that it acquired new functions during the acclimation process and adapted to the new conditions.

Alphaproteobacteria and Gammaproteobacteria constitute the main microbial colonizers of corals[40,47] and *N. vectensis*[18,48]. The increased thermal tolerance of animals acclimated at high temperatures is often associated with an increase in the abundance of these bacterial classes in the associated microbiota[49,50]. In thermally stressed animals, Alphaproteobacteria constitute an important antioxidant army within the coral holobiont[51] and together with members of the Gammaproteobacteria significantly inhibited the growth of coral pathogens (e.g., *V. coralliilyticus* and *V. shiloi*)[7,52]. They are also known to exert nitrogen fixation in endosymbiosis with marine animals, providing the host with additional nutrient supply[53–55]. In our study, Alphaproteobacteria significantly increased in abundance in the animals acclimated at high temperature and most of the bacterial OTUs significantly overrepresented in the animals transplanted with the 25 °C acclimated microbiota belong to the Alpha- and Gammaproteobacteria. Among these OTUs are members of the genera *Sulfitobacter, Francisella*, and *Vibrio*, and one Flavobacteriia OTU of the genera *Muricauda*. All these bacterial groups are known pathogens or symbionts of multicellular organisms[56]. In particular, *Sulfitobacter* is an endosymbiont of vestimentiferans inhabiting hydrothermal vents, where it performs sulfite oxidation[57]; *Francisella* is an intracellular pathogen of mammals and various invertebrates and it is supposedly capable of ROS scavenging[58,59]. Members of the Flavobacteriaceae are key players in biotransformation and nutrient recycling processes in the marine environment, known intracellular symbionts of insects and intracellular parasites of amoebae[60]. All these characteristics make them promising candidates for providing thermal tolerance to the host.

### Changes in host gene expression may confer acclimation.
Acclimation is generally thought to be driven by shifts in gene expression[2,3]. Microbial transplantation experiments allowed us to measure the contribution of bacterial plasticity to host acclimation separately from genetic factors. Although we also observed significant adjustments in the transcriptomic response to thermal acclimation in *N. vectensis*, we were unable to assess the contribution of changes in gene expression to thermal acclimation. The adjustments in gene expression are most likely a combination of acclimation to the new temperature condition and to the changed microbial colonization.

This hypothesis is supported on one hand by the observed adjustments of gene expression involved in innate immune

responses in acclimated animals. In *N. vectensis* polyps acclimated to high temperature, we observed a downregulation of genes involved in innate immunity. Previous studies on *Hydra* showed that the cnidarian innate immune system actively controls the composition and the homeostasis of the associated microbiota and that such associations are both species-specific and life-stage specific[61–64]. Animals challenged by unfavorable environmental conditions (high temperature in this case), may suppress their immune reaction to favor the establishment of new symbionts. In corals, it has been shown that non-acclimated individuals expressed stronger immune and cellular apoptotic responses than acclimated ones, and disease-related metabolic pathways were significantly enhanced in the former[25]. Moreover, the immune system is sensitive to environmental change[42], and colonization by beneficial symbionts might lead to the suppression of the host immune response[38]. Elements of the innate immune system, including several members of the interleukin signaling cascades and the transcription factor NF-kB, have been characterized in *N. vectensis* and are hypothesized to play similar roles as their vertebrate homologs[15,65–67]. Interestingly, a GO term comprising genes implicated in viral processes was upregulated in the animals acclimated at 15 °C, suggesting a possible higher susceptibility of these animals to infections and a possible implication to their lower viability.

On the other hand, the upregulation of genes involved in steroid biosynthesis and metabolism in animals acclimated to high temperature may indicate the contribution of steroid signaling in the regulation of phenotypic plasticity[68,69], e.g., in body size regulation and reproduction rate in response to different temperatures. The enhanced production of small RNAs (sRNAs) in the animals acclimated to high temperature, and the upregulation of processes involved in chromatin remodeling in animals acclimated at 15 °C, suggest a general change in transcriptional and translational regulations at these two extreme conditions, which might take part in acclimation. Chromatin remodeling processes are implicated in epigenetic modifications and thus possibly inheritable by the offspring[70].

A recent publication[71] analyzed coral-associated bacteria proteomes and detected potential host epigenome-modifying proteins in the coral microbiota. This, in concert with specific symbionts inheritance, may constitute an additional mechanism for thermal resistance transmission along with generations and may explain the significantly higher fitness of the 25 °C acclimated animals' offspring.

### Acquired higher fitness is transmitted to the next generation.
The capacity of a species to survive and adapt to unfavorable environmental conditions does not only rely on the adaptability of the adults but also on the survival of the early life stages. Even if the adults can acclimate to periodic heat waves and seasonal temperature increases, their offspring may have a much narrower tolerance range[72–75]. There is evidence that offspring of marine species, including fishes, mussels, echinoderms and corals can acclimatize to warming and acidifying oceans via transgenerational plasticity[76–83]. Both transmission of epigenetic modifications[81,84–88] and microbiota-mediated transgenerational acclimatization[8,23] may be involved in this process.

A recently published work showed that *N. vectensis* polyps acclimated to high temperature transmit thermal resistance to their offspring[89]. In our experiments, we moved a step further by exploring the potential contribution that the microbiota may have in the inheritability of this plasticity. We fertilized oocytes of acclimated females with sperm of a single male to keep the genetic variability as low as possible, and cultured the offspring in a full factorial design at 15, 20, and 25 °C. As expected, offspring

originating from females acclimated at 25 °C showed the highest survival rate. These results confirmed that polyps acclimated to high temperature transmit a higher overall fitness to their offspring. The fact that offspring from genetically identical female polyps show differences in overall fitness suggests either the vertical transmission of specific beneficial bacteria, the transmission of epigenetic modifications, or a combination of both.

For many marine invertebrates, vertical transmission of microbial symbionts is assumed[90–93]. In particular, species that undertake internal fertilization and brood larvae, tend to preferably transmit their symbionts vertically, whereas broadcast spawners and species that rely on external fertilization are thought to mainly acquire their symbionts horizontally[94–96]. Bacteria may also be transmitted to the gametes by incorporation into the mucus that surrounds oocyte and sperm bundles[97–99]. Alternatively, the gametes may acquire bacteria immediately after release by horizontal transmission through the water, which contains bacteria released by the parents[38]. A recent publication showed that *N. vectensis* adopts a mixed-mode of symbiont transmission to the next generation, consisting of a differential vertical transmission from male and female parent polyps, plus a horizontal acquisition from the surrounding medium during development[19]. Consistent with the results of this study, our results suggest vertical transmission of heat tolerance-promoting bacteria.

### Acclimated microbiota, a source for assisted evolution.
Microbial engineering is nowadays regularly applied to agriculture and medicine to improve crop yields and human health[100]. Pioneering theoretical works, including the Coral Probiotic Hypothesis[7] and the Beneficial Microorganisms for Corals concept[101], suggested that artificial selection on the microbiota could improve host fitness over time frames short enough to cope with the actual and future rates of climate changes. Some studies have started microbial engineering on corals as a restoration/conservation option for coral reefs subjected to environmental stresses[102–104]. Recently, corals exposed to experimental warming and inoculated with consortia of potentially beneficial bacteria were shown to bleach less when compared to corals that did not receive probiotics[105]. It needs to be pointed out that microbiota-mediated transgenerational acclimatization[8,23] is of pivotal interest because it would be a suitable target for manipulations in the perspective of future assisted-evolution programs[23,105].

In this study, we proved that long-term acclimation induces enormous changes in the physiology, ecology and even morphology of genetically identical animals. Animals exposed to high temperatures can acclimate and resist heat stress, and this resistance can be transmitted to non-acclimated animals by microbiota transplantation and most likely also to the next generations. We were able to pinpoint specific bacterial groups that may be responsible for different thermal tolerances of their hosts and may be good candidates for future assisted-evolution experiments.

## Methods

**Animal culture**. All experiments were carried out with polyps of *N. vectensis* (Stephenson 1935). The adult animals of the laboratory culture were F1 offspring of CH2XCH6 individuals collected from the Rhode River in Maryland, USA[13,17]. They were kept under constant, artificial conditions without substrate or light in plastic boxes filled with ca. 1 L *Nematostella* Medium (NM), which was adjusted to 16‰ salinity with Red Sea Salt® and Millipore $H_2O$. Polyps were fed two times a week with first instar nauplius larvae of *Artemia salina* as prey (Ocean Nutrition Micro *Artemia* Cysts 430−500 gr, Coralsands, Wiesbaden, Germany) and washed once a week with media pre-incubated at the relative culture temperatures. No individual animals were used for more than one experiment of this study. The animals that survived/were left from each experiment were immediately sacrificed.

**Animal acclimation**. A single female polyp from the standard laboratory culture conditions (16‰, 20 °C) was isolated and propagated via clonal reproduction. When a total of 150 new clones was reached, they were split into 15 different boxes with 10 animals each. The boxes were moved into three different incubators (five boxes each) set at three different acclimation temperatures (ATs) (15, 20, and 25 °C). The animals were kept under a constant culture regime as described above. When the total of 50 polyps per box was reached, it was maintained constant by manually removing the new clones formed. Every week the number of new clones, dead, and spontaneous spawning events were recorded.

**Dry weights**. Ten animals from each AT were rinsed quickly in pure ethanol and placed singularly in 1.5 ml tubes, previously weighed on an analytical scale. The animals were left dry at 80 °C in a ventilated incubator for 4 h. After complete evaporation of fluids, the animals with the tubes were weighed on the same analytical scale and the dry weight was calculated.

**Generation of axenic polyps**. To reduce the total bacterial load (axenic state), animals belonging to the same clonal line were treated with an antibiotic cocktail of ampicillin, neomycin, rifampicin, spectinomycin, and streptomycin (50 μg/ml each) in filtered (on 0.2 μm filter membrane), autoclaved NM (modified from ref. [106]). The polyps were incubated in the antibiotic cocktail for 2 weeks in 50 ml Falcon tubes (10 animals each). The medium and the antibiotics were changed every day and the tubes three times per week. After the treatment, the polyps were incubated for 1 week in sterile NM without antibiotics to let them recover before the recolonization. After the 2 weeks of antibiotic treatment, the axenic state was checked by homogenizing single polyps with an electric tissue grinder (Omni THq Homogenizer) into 1 ml sterile NM and by plating 100 μl of the homogenate on marine broth plates, successively incubated for 1 week at 20 °C. In addition, we performed a PCR with primers specific for the V1-V2 region of the bacterial 16S rRNA gene (27F and 338R). No CFUs on the plates and a weaker signal in the PCR electrophoretic gel compared with wild-type controls were considered evidence of bacterial reduction and an axenic state of the animals.

**Heat stress experiment (HS)**. Adult polyps from each AT were placed singularly in six-well plates and incubated at 40 °C for 6 h (adapted from[72]). The day after, the number of survivors was recorded and the mortality rate was calculated.

**Bacterial transplantation**. For this experiment, the protocol for conventionalized *Hydra* polyps was modified[106]. For each acclimated culture ($n = 5$), 6 axenic adult polyps were recolonized with the supernatant of 1 adult polyp, singularly homogenized (as described above) in 2 ml sterile NM. The solution was centrifuged for 5 s at $1 \times g$ to sediment the coarsest tissue debris. One ml of supernatant was added into single Falcon tubes, containing 6 axenic animals each and filled with 50 ml sterile NM. One additional animal from each acclimated culture was collected for DNA extraction and 16S sequencing (donors). The tubes with the recolonized animals were placed at 20 °C and after 24 h, the medium was exchanged to remove tissue debris and non-associated bacteria. One month after recolonization, three recolonized animals of each replicate were tested for heat stress tolerance as described above ($n = 15$) and three recolonized polyps of each replicate were sampled for DNA extraction and 16S sequencing.

**Induction of spawning**. Acclimated animals separated singularly in six-well plates, were induced for sexual reproduction via light exposure for 10 h and temperature shift to 20 °C for the animals acclimated at 15 °C[17]. The animals acclimated at 20 and 25 °C were shifted to 25 °C. At each fertilization event, sperms from a single induced male were pipetted directly onto each oocyte pack. Fertilization was performed within 3 h after spawning. The developing animals were then cultured for 1 month under different developmental temperatures (DTs) (15, 20, or 25 °C).

**Offspring survival test**. Two female polyps from each of the acclimated cultures ($n = 5$) and one male polyp from the standard culture conditions were induced separately for spawning. After spawning, the adult polyps were removed and the oocyte packs fertilized as described above. Fertilization was confirmed by observation under a binocular of the oocytes' first cleavages. After fertilization, each oocyte pack was split with a scalpel into 3 parts that were transferred into 3 distinct Petri dishes. The 3 oocyte-pack sub-portions were placed into 3 different incubators, set at 15, 20, and 25 °C, respectively, and let develop for 1 month. Right after fertilization and after 1 month of development, pictures of the oocytes and the juvenile polyps were acquired for successive counting. Ratios between the initial number of oocytes and the surviving juvenile polyps were calculated and the survival rate was estimated.

**Bacteria vertical transmission test**. One female polyp from each of the acclimated cultures ($n = 5$) and one male polyp from the standard conditions were induced separately for spawning as described above. Immediately after spawning the parental polyps were collected, frozen in liquid nitrogen and stored at −80 °C for successive DNA extraction. Oocyte packs were fertilized, split into three parts each and let develop for 1 month at the three different DTs, as described for the

offspring survival test. After 1 month of development, the juvenile polyps were collected, frozen in liquid nitrogen and stored at −80 °C. DNA was extracted from both the adults and the offspring as described herein.

**DNA extraction**. DNA was extracted from adult polyps starving for 3 days before sacrifice and from never-fed juveniles. The recolonized animals were not fed for the whole duration of the antibiotic treatment and the transplantation (7 weeks in total). Animals were washed two times with 2 ml autoclaved MQ, instantly frozen in liquid nitrogen without liquid, and stored at −80 °C until extraction. The gDNA was extracted from whole animals with the DNeasy®Blood & Tissue Kit (Qiagen, Hilden, Germany), as described in the manufacturer's protocol. Elution was done in 50 μl and the eluate was stored at −80 °C until sequencing. DNA concentration was measured by gel electrophoresis (5 μl sample on 1.2% agarose) and by spectrophotometry through Nanodrop 3300 (Thermo Fisher Scientific).

**RNA extraction**. Adult animals were starved for 3 days before sacrifice. Polyps were washed two times with 2 ml autoclaved MQ, instantly frozen in liquid nitrogen without liquid and stored at −80 °C until extraction. Total RNA was extracted from the body column only, with the AllPrep® DNA/RNA/miRNA Universal Kit (Qiagen, Hilden, Germany), as described in the manufacturer's protocol. RNA elution was done in 20 μl of RNase-free water and the eluates were stored at −80 °C until sequencing. RNA concentration was measured through electrophoresis by loading 1 μl of each sample on 1% agarose gel and by spectrophotometry through Nanodrop 3300 (Thermo Fisher Scientific).

**16S RNA sequencing and analysis**. For each DNA sample, the hypervariable regions V1 and V2 of bacterial 16S rRNA genes were amplified. The forward primer (5′- **AATGATACGGCGACCACCGAGATCTACAC** XXXXXXXX TATGGTAATTGT AGAGTTTGATCCTGGCTCAG-3′) and reverse primer (5′- **CAAGCAGAAGACGGCATACGAGAT** XXXXXXXX AGTCAGTCAGCC TGCTGCCTCCCGTAGGAGT-3′) contained the Illumina Adaptor (in bold) p5 (forward) and p7 (reverse)[107]. Both primers contain a unique 8 base index (index; designated as XXXXXXXX) to tag each PCR product. For the PCR, 100 ng of template DNA (measured with Qubit) were added to 25 μl PCR reactions, which were performed using Phusion® Hot Start II DNA Polymerase (Finnzymes, Espoo, Finland). All dilutions were carried out using certified DNA-free PCR water (JT Baker). PCRs were conducted with the following cycling conditions (98 °C—30 s, 30 × [98 °C—9 s, 55 °C—60 s, 72 °C—90 s], 72 °C—10 min) and checked on a 1.5% agarose gel. The concentration of the amplicons was estimated using a Gel Doc TM XR + System coupled with Image Lab TM Software (BioRad, Hercules, CA USA) with 3 μl of O'GeneRulerTM 100 bp Plus DNA Ladder (Thermo Fisher Scientific, Inc., Waltham, MA, USA) as the internal standard for band intensity measurement. The samples of individual gels were pooled into approximately equimolar subpools as indicated by band intensity and measured with the Qubit dsDNA br Assay Kit (Life Technologies GmbH, Darmstadt, Germany). Sub-pools were mixed in an equimolar fashion and stored at −20 °C until sequencing. Sequencing was performed on the Illumina MiSeq platform with v3 chemistry ($2 \times 300$ cycle kit)[108]. The raw data are deposited at the Sequence Read Archive (SRA) and available under the project ID PRJNA742683.

The 16S rRNA gene amplicon sequence analysis was conducted through the QIIME 1.9.0 package[109]. Sequences with at least 97% identity were grouped into OTUs and clustered against the QIIME reference sequence collection; any reads, which did not hit the references, were clustered de novo. OTUs with less than 50 reads were removed from the dataset to avoid false positives which rely on the overall error rate of the sequencing method[110]. Samples with less than 3,600 sequences were also removed from the dataset, being considered as outliers. For the successive analysis, the number of OTUs per sample was normalized to that of the sample with the lowest number of reads after filtering.

Alpha-diversity was calculated using the Chao1 metric implemented in QIIME. Data were subjected to descriptive analysis, and normality and homogeneity tests as described herein. When normality, homogeneity and absence of significant outliers assumptions were met, statistical significance was tested through one-way ANOVA. When at least one of the assumptions was violated, the non-parametric Kruskal–Wallis test was performed instead. When a significant difference between treatments was detected, post-hoc comparisons were performed to infer its direction and effect size. Tukey's post hoc comparisons were applied after ANOVA, while Dunn's post hoc after Kruskal–Wallis.

Beta-diversity was calculated in QIIME according to the different β-diversity metrics available (Binary-Pearson, Bray–Curtis, Pearson, Weighted-Unifrac, and Unweighted-Unifrac). Statistical values of clustering were calculated using the non-parametric comparing categories methods Adonis and Anosim.

Bacterial groups associated with specific conditions were identified by LEfSe (http://huttenhower.sph.harvard.edu/galaxy)[111]. LEfSe uses the non-parametric factorial Kruskal–Wallis sum-rank test to detect features with significant differential abundance, concerning the biological conditions of interest; subsequently, LEfSe uses Linear Discriminant Analysis (LDA) to estimate the effect size of each differentially abundant feature. In addition to that, presence-absence calculations were performed directly on the OTU tables to detect bacterial OTUs that are shared between donor and recipient and between F0 and F1 bacterial

communities. Statistical tests were performed through JASP v0.16 (https://jasp-stats.org).

**Transcriptome analyses**. The analysis was performed on five animals from each AT in two repeated sequencing runs. mRNA sequencing with previous poly-A selection was performed for 15 libraries on the Illumina HiSeq 4000 platform, with 75 bp and 150 bp paired-end sequencing, respectively. The quality of raw reads was assessed using FastQC v0.11.7 (Andrews, 2014[112]). Trimmomatic v.0.38[113] was then applied to remove adaptors and low-quality bases whose quality scores were less than 20. Reads shorter than 50 bp were removed, and only paired-end reads after trimming were retained. Reads were mapped to the Ensembl metazoa *Nematostella vectensis* genome (release 40) using the splice-aware aligner hisat2 v2.1.0[114] with rna-strandness RF option and default parameters (Supplementary Table 7).

RNA-seq data was used to improve the predicted *N. vectensis* gene model downloaded from Ensembl Metazoa database release 40. Using mapped reads from each temperature condition as input, StringTie v2.0[115] and Scallop v0.10.4[116] were applied to perform genome-guided transcriptome assemblies. The assembled transcripts were subsequently compared and merged using TACO[117]. This produced 42,488 genes with 81,163 transcripts, among which 21,245 genes had significant matches (blastx with parameter $e$-value $1^{e-5}$) with proteins in the SwissProt database. Assembled genes were compared with the Ensembl gene model using gffCompare v0.11.2[118], from which genes with lower blastx $e$-value were selected. Ensembl genes without matching assembled genes were retained, and assembled genes without matching Ensembl genes but with significant matching SwissProt proteins were added to the gene model. The final gene model included 20,376 Ensembl genes, 4,400 improved genes and 2,751 novel assembled genes (Supplementary Data 4). The gene model statistics and the completeness of gene models were assessed using BUSCO v5.2.2[119] on the Metazoa dataset (Supplementary Table 8, Supplementary Table 9). Total counts of read fragments aligned to the annotated gene regions were derived using the FeatureCounts program (Subread-2.0.0)[120] with default parameters. Genes whose combined counts from all samples were lower than 5 counts per million (cpm) mapped reads were excluded from the analyses. Differential expression analyses were performed in parallel using DESeq2 (v1.28.1) BioConductor package[121], and limma (voom v3.44.3) package[122]. Differentially expressed genes (DEGs, Supplementary Table 10) were determined based on false-discovery rate (FDR, Benjamini–Hochberg adjusted $p$-value ≤ 0.05). Gene ontology annotation was derived from the best-matching SwissProt proteins. Enriched GO-terms in DEGs were identified by the topGO (v2.40.0) BioConductor package (Supplementary Data 1).

**Reporting summary**. Further information on research design is available in the Nature Research Reporting Summary linked to this article.

## Data availability

Source data and supporting information files are provided with this paper. Transcriptomic data and 16S rRNA gene sequencing data are available at the NCBI database under accession codes GSE168938 and PRJNA742683. Source data are provided with this paper.

## Code availability

No custom code was used in this study.

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

## Acknowledgements

This work was supported by the Human Frontier Science Program (Young Investigators' Grant RGY0079/2016) to SFrau and AMR and the DFG CRC grant 1182 "Origin and Function of Metaorganisms" (Project B1) to SFrau. Sequencing was supported by the DFG Research Infrastructure NGS_CC (project 407495230) as part of the Next Generation Sequencing Competence Network (project 423957469). N.G.S. was carried out at the Competence Centre for Genomic Analysis (Kiel). We thank Katja Cloppenborg-Schmidt for preparing the 16S rRNA gene library (CRC1182, Z3).

## Author contributions

L.B., H.Y., A.R., and S. Fraune designed experiments; L.B. performed experiments; L.B., H.Y., and S. Fraune analyzed data; S. Franzenburg provided RNA sequencing, L.B. and S.F. wrote the paper, L.B., H.Y., A.R., and S. Fraune revised the manuscript.

## Funding

## Competing interests

The authors declare no competing interests.
