## [Peer Review File · Nature Communications]

REVIEWER COMMENTS

Reviewer #1 (Remarks to the Author):

General comments:

The study by Baldassarre et al reports on a long-term temperature acclimation experiment looking at the potential contribution of the microbiome to the temperature resilience of the sea anemone *Nematostella vectensis*. The authors incubated clonal individuals for over 3 years at three different temperatures and analyzed their response to a 40C heat stress experiment, the associated changes in their microbiome structure as well as the physiological performance on the level of growth and fecundity. The authors further transferred the microbiomes to naïve polyps and tested the effects of this transplantation on their thermal resilience. Interestingly, the authors find that naïve polyps receiving the microbiomes from the high temperature treated donors showed significantly higher survival rates, suggestion that transfer of the microbiome from previously acclimated individuals is sufficient to promote increases in temperature resilience. In order to determine if the microbiome is also naturally transferred to sexually produced offspring, the authors performed multiple crossing experiments. Analysis of the microbiome of offspring from the different treatments subjected to different temperature regimes during development suggest that a significant portion of the microbiome is indeed transferred to the next generation. The authors conclude that the microbiome can indeed confer an increase in temperature resilience and that this can be transferred to the next generation via vertical transmission of the microbiome from the mother. These findings are further put in context in light of climate change and the potential contribution of the microbiome in the acclimation/adaptation of vulnerable organisms to increasing temperatures.

The manuscript is language wise acceptable, but another round of proofreading is required to remove the typos and grammatical errors that are still present. Regarding the data provided, and their interpretation and conclusions, I do have several points that require clarification and I believe that some of the conclusions are not justified and require some toning down. Specifically, I'd like to point out that the authors cannot fully disentangle the contribution of the microbiome and other acclimation responses for most of the experiments. Having said that, I do like the microbiome transplantation experiment as naïve polyps were used that were not exposed to any of the temperature treatments. I believe that this is, by far, the strongest part of the study and most novel contribution. However, other experiment, such as those testing the contribution of the microbiome to the increased resilience of offspring from the parental generation are flawed in their ability to disentangle microbiome contributions from potential epigenetic mechanisms. The authors are aware of this as they specifically mention it in the discussion, however, the wording used in the different sections throughout the manuscript does not reflect this (see specific comments below). I urge the authors to tone down these respective sections and to not oversell their findings. Similar issues apply to the RNAseq experiment as well as the microbiome inheritance experiments as these could also be mainly driven by temperature i.e. they show the similarities because the organisms were exposed to the same treatment. Most importantly, the limitations of the experiments and the data provided also needs to be reflected in the overall conclusions presented in the abstract i.e. "Microbial community analyses of the F1 generation revealed the transmission of the acclimated microbiota to the next generation. These results indicate

that microbiota plasticity can contribute to animal thermal acclimation and its transmission to the next generation may represent a rapid mechanism for thermal adaptation.” The data provided does not convincingly show that the microbiome is vertically transmitted and that this transmission represents a mechanism of rapid adaptation. We agree that the authors used the word “may” but I still feel that the wording is too strong in the context of the caveats of the study. Also, the temperature stress experiment performed at 40C is rather a heat shock, which is somewhat different from a normal heat stress event where temperature increase slower and less extreme with regard to the overall temperature differences e.g. 15C acclimated animals exposed to 40C without any ramping. This should be clearly stated in the respective sections.

Furthermore, we suggest checking the references as not all list the journal where they were published. The format of the figures, figure legends, and referencing figures in text, need to be adjusted to match the format of Nature Communications.

Do not add a space between the word and the reference.

Specific comments:

Lines 30 – 32: Multicellular organisms have already faced increased temperatures on earth and are still around. The authors should rephrase it to something as “the increased temperatures pose a threat to the survival of organisms” or so on.

Line 32: remove alo.

Line 45: the current data presented doesn't show that the microbiome is indeed passed on as the plots do not compare both donor and recipient microbiome profiles

Line 64: I am not sure that the concept of symbiosis mediated evolution of species fits here as it substantially differs from microbiome mediated acclimation from a mechanistic point of view

Line 72: sentence broken?

Line 76: change borrowed for burrowed.

Line 85: Recently, has been shown that female?

Line 98: change evidences to evidence.

Line 316: remove the extra space before experiment.

Line 229: update analysis to BUSCO 5 and to also include the number of duplicated genes

Line 320: change acclimated to acclimated.

Line 328: change acclimation temperature to acclimation temperatures

Line 329: change where to were.

Figure 2B: are these deaths per box or as a total of all boxes? If it is per box, then change week*box to week/box. Same for panel C.

Figure 2: wrong panel letter, change (E) to (C).

Line 356: remove rate at 25°C.

Line 375: move (Figure 1) after woa.

Figure 2: panel 3b mentions 70 woa instead of 84.

Line 381: change pinary-pearson to Binary-pearson

Line 388: change was to were.

Lines 410-411: would be good to include the statistical analysis as a supplementary table.

Line 451: wrong table referred to, should be S5, not S3

Line 469: would be good to add the data of the 16S rRNA sequencing from the donors to the PCoA of the transplanted microbiota, and also compare their microbial compositions to see if there were changes in microbial communities due to transplantation efficiency and microbiome re-acclimation to 20°C. This could go to supplementary information or included as an extra panel in figure 5. Additionally, was there any difference in size in the polyps transplanted with the microbiome from 15, 20, or 25°C after the month of acclimation at 20°C?

Line 488: Doesn't really show that the microbiome has been transferred as no comparison of donor and recipient microbiomes are shown

Line 505: change "to next generation" to "to the next generation"

Line 505 (and 544): header suggests that microbiome and heat tolerance are transferred to the next generation but no comparison of donor recipient microbiomes is shown, and, heat tolerance was not assessed as for the previous microbiome transfer experiment. Wording too strong and conclusion is not justified by the data provided.

Line 544: the authors mention that "the fact that also parts of the acclimated microbiota is transmitted and persisting in the juvenile F1 polyps..."; however, they never show the microbiome composition of the juvenile F1 polyps. These data are required and could be added as an extra panel in Figure 6. Moreover, a statistical analysis of the differences in bacterial OTUs abundances, such as the one

presented in Table S6, is required.

Line 544: Was the size of the F1 polyps also measured? I would assume AT 25°C derived polyps to be larger but is good information to have.

Line 566: There was bleaching, just lower than not acclimated corals.

Line 572: This study does not disentangle the contribution of host gene expression and microbiota to temperature acclimation in cnidarians, as only one time point was collected for RNASeq data and it wasn't even under heat stress. It does, however, provide gene expression information of *N. vectensis* polyps acclimated to different temperatures at 75 woa and with different microbiota. It also provides insights into the effects of microbiome transplantation on heat tolerance in *N. vectensis*.

Line 594: Similar to the previous comment. This isn't disentangling gene expression and microbiota contribution, as gene expression is likely going to be different with different microbiota. But does remove the genotype effect by working with clones. More importantly, it does show the big impact of microbiota on survival.

Line 620: The paper that is cited does mention that there is an increased microbiota β -diversity in corals upon heat stress but argues the opposite to the point brought by the authors. Where increased β -diversity was associated with coral mortality, not tolerance. Additionally, the citation is given to a paper that cites another, the proper citation should be to: Zaneveld, J. R. et al. Overfishing and nutrient pollution interact with temperature to disrupt coral reefs down to microbial scales. *Nat. Commun.* 7, 11833 (2016).

Line 674: "and antifouling activity reference and reference found ..." Please check this sentence.

Line 709: The results show an increase in survival at every temperature in offspring derived from the acclimated 25°C females (Fig. 6b). This can be observed at 15, 20, and 25°C. This suggests for an overall increase in survival of the offspring derived from 25°C acclimated mothers and not a thermal specific tolerance.

Reviewer #2 (Remarks to the Author):

The manuscript entitled "Microbiota mediated plasticity promotes thermal adaptation in *Nematostella vectensis*" represents an impressively comprehensive long-term project studying the effects of temperature acclimation on a cnidarian holobiont. I found the experimental design appropriate for the study question and the data analysis suitable for the claims. My suggestions for minor revisions for clarification are detailed below.

Methods:

Line 144: Clarify “smashing”. Is this homogenization? If so- what homogenization procedure was used? (e.g., mortar and pestle; bead beater; syringe; Dounce homogenizer; electric tissue grinder)

Line 154: Clarify what is the supernatant of acclimated adult polyps. The supernatant is from the anemone “smashing”? Please provide more detail here about the homogenization procedure and how the supernatant was obtained (centrifugation?).

Lines 171–173; 179–180; and 192–193: Are the spawning animals the same animals that just came out of the heat stress or did these animals never enter the 40°C heat stress? Please clarify.

Line 195: Please clarify what “Five not induced female polyps” means in this sentence. Five females that did not enter the spawning cycle?

Line 198: The abbreviation “DT” is used here and it seems like it could be have been introduced in line 176 for the other developing temperature earlier.

Figures:

Figure 1: This figure is greatly appreciated given the impressive long-term experimental design! I would find it helpful if there were more explanations for the sample size at each step. For example, “HS (1)” could be “HS 1 (n = x)”.

Figure 3: I might recommend combining panels A and B by using different shapes for 40, 70, and 132 weeks while keeping the colors for the different ATs. Either way the point is well illustrated but I did have trouble seeing the details of the figures because there is so much going on in Figure 3 (a good problem!).

Figure 5: Missing a closing parenthesis in the figure legend.

Discussion:

Line 600: The authors make a good point about the animals being the same age and size. You observed considerable variation in size and cloning rate between the different acclimation temperatures. If you made an effort to track single polyps throughout time to make sure they were all the same age, please note that here. Otherwise it seems that some polyps of the original 150 must have been older than others given that they originated from a single individual.

Minor edits:

Line 32: Typo: “... through genetic recombination and natural selection alo.” Should be alone?

Line 86: Typo: “Recently, has been shown that...” Missing a word?

Line 109: Typo. No period between sentences ending in “USA” and starting with “They”

Lines 135–139: This sentence is difficult to follow. I think there are unnecessary words that are making it difficult to understand (e.g., “and remove the most of associated bacteria”).

Line 200 and 207 and 217: Is “N” short for nitrogen? Saying nitrogen may be more clear or the chemical abbreviation LN2.

Line 216: Typo: “Adult animals starved for 3 days...” missing the word “were”?

Line 332: Typo: “...where performed at 40 and 132 weeks...” should be “were performed”

Line 589: subscript 2 in carbon dioxide chemical abbreviation

Line 595: “microbiotas” should be “microbiota” or “microbes”

Reviewer #3 (Remarks to the Author):

In this study the authors examine the effects of temperature acclimation, and the associated effects of the microbiome, on the survival of *N. vectensis* at higher temperatures. They also examine the transplantation of the microbiota and examine those effects. This study was well thought out, with appropriate controls and well-made figures. I especially appreciated the diagrams explaining the experimental procedures in Figures 1 and 6. The study incorporates not only 16S data, but also transcriptome data. It was a pleasure to read. The long-term duration of this experiment was also successful in showing how long it took for the microbiome to become stable in the different temperature conditions. The authors clearly demonstrate that the microbiome of *N. vectensis* acclimated at higher temperatures long-term are better equipped to cope with temperature stress. Work linking cnidarian adaptation to microbiome to changing temperatures is much needed, and long-term experiments of this sort are rare. My comments are all fairly minor, and mostly have to do with clarity. I think this is a really well done paper, and strongly support publication. Great job!

Comments:

- 1) In line 56 you refer to the Modern Synthesis. While this is a concept that most should be familiar with, my quick poll of some colleagues (especially those on the younger side) suggests that maybe it is less well known (at least by this term) than it should be. I'd suggest citing the original work.
- 2) You are using a lot of acronyms in this manuscript, and maybe don't need all of them. I would probably try to spell out woa and AT in each figure legend, and other acronyms like HTPB are only used a couple of times and could just be spelled out to avoid confusion. Same with ME, BMC, AE, MMTA.
- 3) In Figure 2 I think it was a little unclear if the sample in B/C/D were being held at the AT throughout – I might include that info in the legend. (I was able to figure it out from the text, but I think it's good for the figures to be able to stand on their own.)
- 4) In Figure 3, I both like and dislike that the PCoA was shown twice with different grouping. I think you're making an important point and having them shown twice is useful for that, but I think it could be worthwhile to indicate in some way that the two figures show the same samples. At a minimum I'd state this in the legend, but it might be nice to indicate woa with shape in A, and AT with shape in B, or something like that. I think the same comment applies for Figure 6 C and D. (I liked that you were consistent with color choice throughout, so not suggesting you change that!)
- 5) The one result I found surprising was the low survival of polyps acclimated to 25°C after heat stress, which resulted in many more dead polyps after 40 weeks, although not at 132 weeks. You mention a possible explanation for this in lines 560-561 (although I think you should clarify in those lines that you are discussing the 40°C heat stress). I was thinking that the transplantation experiments would support this experiment as well, but I don't think you did the strong heat shock on those samples. Looking more

closely at Figure 3, I'm still struggling with this a bit, because at the 40 woa, the microbiome of the 25 and 20 AT still cluster together fairly closely, so it doesn't seem like the microbiome is distinct enough to explain such a strong change in survival rates. I'd appreciate more discussion of this point in the discussion.

Minor comments:

Line 32: ...natural selection also...

Line 34: ... microbiota as a putative source...

Line 56: I felt that

Line 76: burrowed, rather than borrowed

Line 121: maybe "the polyp number was maintained consistently"

Line 123: were, rather than where

Line 320: acclimated, rather than acclimated

Line 328: temperatures, rather than temperature

Line 347: were, rather than was

Line 357: I think the temperature here should be 40°, not 25° "the mortality rate at 25°C was significantly reduced"

Line 348: I think it should say "per box" rather than "and box"

Line 348: Refers to panel E which isn't present – I think it should be C

Line 381: binary, rather than pinary

Line 437-438: ...showing the number of differentially expressed genes within the three AT...

Figure 6: The AT designation in B looks like it belongs to 20°C only. It might be clearer to put AT above those three temperatures.

Line 573: cnidarians, rather than cnidarian

Line 597: before, rather than until

Line 609: I found the phrase "punctual abundances" to be hard to interpret here – I would revise to clarify your meaning.

Line 611: ...from the surrounding environment, or a higher...

Line 734-736: I'd revise this sentence to maybe something like: "Recently, corals subjected to experimental warming and inoculated with consortia of potentially beneficial bacteria were shown to bleach less when compared to corals that received no probiotics." (otherwise need to change the beginning as you are missing a noun)

Line 740: I would change the semicolon to a comma, and add an additional comma later in the sentence: ...heat stress, and that this resistance...

We thank the three anonymous reviewers for their insightful and positive comments.

Reviewer #1 (Remarks to the Author):

General comments:

The study by Baldassarre et al reports on a long-term temperature acclimation experiment looking at the potential contribution of the microbiome to the temperature resilience of the sea anemone *Nematostella vectensis*. The authors incubated clonal individuals for over 3 years at three different temperatures and analyzed their response to a 40C heat stress experiment, the associated changes in their microbiome structure as well as the physiological performance on the level of growth and fecundity. The authors further transferred the microbiomes to naive polyps and tested the effects of this transplantation on their thermal resilience. Interestingly, the authors find that naive polyps receiving the microbiomes from the high temperature treated donors showed significantly higher survival rates, suggestion that transfer of the microbiome from previously acclimated individuals is sufficient to promote increases in temperature resilience. In order to determine if the microbiome is also naturally transferred to sexually produced offspring, the authors performed multiple crossing experiments. Analysis of the microbiome of offspring from the different treatments subjected to different temperature regimes during development suggest that a significant portion of the microbiome is indeed transferred to the next generation. The authors conclude that the microbiome can indeed confer an increase in temperature resilience and that this can be transferred to the next generation via vertical transmission of the microbiome from the mother. These findings are further put in context in light of climate change and the potential contribution of the microbiome in the acclimation/adaptation of vulnerable organisms to increasing temperatures.

The manuscript is language wise acceptable, but another round of proofreading is required to remove the typos and grammatical errors that are still present. Regarding the data provided, and their interpretation and conclusions I do have several points that require clarification and I believe that some of the conclusions are not justified and require some toning down. Specifically, I'd like to point out that the authors cannot fully disentangle the contribution of the microbiome and other acclimation responses for most of the experiments. Having said that, I do like the microbiome transplantation experiment as naive polyps were used that were not exposed to any of the temperature treatments. I believe that this is, by far, the strongest part of the study and most novel contribution. However, other experiment, such as those testing the contribution of the microbiome to the increased resilience of offspring from the parental generation are flawed in their ability to disentangle microbiome contributions from potential epigenetic mechanisms. The authors are aware of this as they specifically mention it in the discussion, however, the wording used in the different sections throughout the manuscript does not reflect this (see specific comments below). I urge the authors to tone down these respective sections and to not oversell their findings. Similar issues apply to the RNAseq experiment as well as the microbiome inheritance experiments as these could also be mainly driven by temperature i.e. they show the similarities because the organisms were exposed to the same treatment. Most importantly, the limitations of the experiments and the data provided also needs to be reflected in the overall conclusions presented in the abstract i.e. "Microbial community analyses of the F1 generation revealed the transmission of the acclimated microbiota to the next generation. These results indicate that microbiota plasticity can contribute to animal thermal acclimation and its transmission to the next generation may represent a rapid mechanism for thermal adaptation." The data provided does not convincingly show that the microbiome is vertically transmitted and that this transmission represents a mechanism of rapid adaptation. We agree that the authors used the word "may" but I still feel that the wording is too strong in the context of the caveats of the study. Also, the temperature stress experiment performed at 40C is rather a heat shock, which is somewhat different from a normal heat stress event where temperature increase slower and less extreme with regard to the overall temperature differences e.g. 15C acclimated animals exposed to 40C without any ramping. This should be clearly stated in the respective sections.

Furthermore, we suggest checking the references as not all list the journal where they were published. The format of the figures, figure legends, and referencing figures in text, need to be adjusted to match the format of Nature Communications.

Do not add a space between the word and the reference.

Answer:

We thank the reviewer for his/her comments and suggestions.

- *We agree on the point that with our experiments the contribution of the microbiome to the increased resilience of offspring cannot be separated from epigenetic effects. We are currently addressing this question in a follow-up study. To address this point more carefully, we toned down the respective sections. In addition, we discussed equally both potential vertical transmission of thermal resistance by the microbiota and the potential contribution of epigenetic modifications. Nevertheless, to support our hypothesis of vertical transmitted acclimated bacteria, we included presence/absence analyses of bacterial OTUs (new Table S11), providing strong evidence for vertical transmission of acclimated bacteria. In addition, we provided a new PCoA (new Figure S3) that shows in detail the clustering of F0 and F1 samples.*
- *Similar analyses we have included for the microbial transplantation experiment (new Figure S2, Table S8) showing the clustering of donor and recipient microbiota.*
- *We agree that transcriptome analysis of the acclimated animals stresses the fact that host gene regulation may also plays a role in the thermal acclimation process. A GO term analysis revealed a high proportion of DE genes potentially involved in epigenetic modifications supporting the reviewer's point of view that epigenetic remodelling is likely also taking part in the thermal acclimation process. We agree that the microbiome is most likely only one factor contributing to fast acclimation and that epigenetic modification most likely is a second factor involved in thermal acclimation and adaptations. We discussed that point in the discussion (line 750-753).*
- *We maintained our main conclusions, as we are convinced that our additional analyses justify them. However, we tone down parts of the abstract.*
- *Finally, we would like to point out that although exposure of animals to 40 °C for 6 h without ramping seems to be comparable to heat shock, we designed our experiment based on previously published protocols for *Nematostella* (Reitzel et al., 2013) and natural temperature conditions for *Nematostella*. *Nematostella* lives in the intertidal zone, where water temperature of above 40°C can quickly be reached in small tidal pools*
- *We have proofread the manuscript once again and removed further mistakes.*

Specific comments:

Lines 30 – 32: Multicellular organisms have already faced increased temperatures on earth and are still around. The authors should rephrase it to something as “the increased temperatures pose a threat to the survival of organisms” or so on.

Answer: *done.*

Line 32: remove *alo*.

Answer: *we corrected to “alone”.*

Line 45: the current data presented doesn't show that the microbiome is indeed passed on as the plots do not compare both donor and recipient microbiome profiles

Answer: *To show convincingly that acclimated bacteria were successfully transplanted to naive recipients, we now included donor and recipient samples in one plot and added Figure S2 with more information. We show the clear dependence of recipient microbiota from donor microbiota. In addition,*

we performed a presence/absence analysis (new Table S8). Thereby, we compared OTU counts of donor communities with that of recipient communities and identified OTUs present in both datasets. These comparisons demonstrate the successful transmission of bacteria on OTU level and the differential presence of OTUs specific for the different acclimation treatments in the recipient polyps.

In the main text, we extended the corresponding paragraph (line 480-492):

“To evaluate the rate of vertical bacterial transmission we included the donor samples into the PCoA (Figure S2, Table S7). While principal component 1 explained the bacterial variation due to transplantation, principal component 3 explained the bacterial variation due to the difference in acclimation temperature in donor polyps (Figure S2 a-c). In addition, presence-absence analysis based on the OTU read table revealed that recipient polyps received a high proportion of differentially abundant bacteria from the corresponding donor polyps during transplantation (Table S8).

Nevertheless, not all bacteria could be transplanted with the same efficiency. Recipient polyps showed a reduction of bacterial α -diversity by approximately 30% compared to the donor polyps (Figure S2-d). While most bacterial classes were present in similar proportions in donor and recipient polyps, Epsilonproteobacteria did not appear to be transplantable (Figure S2-e).

.”

Line 64: I am not sure that the concept of symbiosis mediated evolution of species fits here as it substantially differs from microbiome mediated acclimation from a mechanistic point of view

Answer: Thank you for this comment. We agree on this point and removed the corresponding sentence.

Line 72: sentence broken?

Answer: we fixed the sentence.

Line 76: change borrowed for burrowed.

Answer: done.

Line 85: Recently, has been shown that female?

Answer: We changed to “Recently, it has been shown that female and male polyps transmit different bacterial species to the offspring and that additional symbionts are acquired from the environment during development”.

Line 98: change evidences to evidence.

Answer: done

Line 316: remove the extra space before experiment.

Answer: Sorry, but we could not detect the extra space.

Line 229: update analysis to BUSCO 5 and to also include the number of duplicated genes

Answer: We updated the analyses to BUSCO 5 and included the updated number of duplicated genes in Table S3.

Line 320: change acclimated to acclimated.

Answer: done.

Line 328: change acclimation temperature to acclimation temperatures

Answer: done.

Line 329: change where to were.

Answer: done.

Figure 2B: are these deaths per box or as a total of all boxes? If it is per box, then change week*box to week/box. Same for panel C.

Answer: *The indicated numbers are deaths per box per week. You can write it in both ways, “deaths/box/week” or death/(box*week). We changed it, as requested.*

Figure 2: wrong panel letter, change (E) to (C).

Answer: *done.*

Line 356: remove rate at 25°C.

Answer: *done.*

Line 375: move (Figure 1) after woa.

Answer: *done.*

Figure 2: panel 3b mentions 70 woa instead of 84.

Answer: *corrected*

Line 381: change pinary-pearson to Binary-pearson

Answer: *done.*

Line 388: change was to were.

Answer: *done.*

Lines 410-411: would be good to include the statistical analysis as a supplementary table.

Answer: *We agree. The statistical data are now presented in Table S6. We realized that we performed in this case not a Two-way ANOVA, but a One-way ANOVA. We corrected it accordingly.*

Line 451: wrong table referred to, should be S5, not S3

Answer: *corrected.*

Line 469: would be good to add the data of the 16S rRNA sequencing from the donors to the PCoA of the transplanted microbiota, and also compare their microbial compositions to see if there were changes in microbial communities due to transplantation efficiency and microbiome re-acclimation to 20°C. This could go to supplementary information or included as an extra panel in figure 5. Additionally, was there any difference in size in the polyps transplanted with the microbiome from 15, 20, or 25°C after the month of acclimation at 20°C?

Answer: *Good points! We expanded the analyses and added a new Figure S2 with following panels 1. we added the donor samples to the PCoA. 2. We included an alpha-diversity analysis and 3. We added a representation of bacterial groups present in donor and recipient polyps. On one hand, these results indicate the successful transplantation of parts of the donor microbiota, on the other hand, they illustrate that not all bacteria could be transplanted with the same efficiency. This gets evident by the alpha-diversity analysis as well as by the representation of the bacterial families. We added the information in line 478-492 in the main text.*

The sizes of the recipient polyps were not measured, but no evident changes were detected.

Line 488: Doesn't really show that the microbiome has been transferred as no comparison of donor and recipient microbiomes are shown

Answer: *We subjected the recipient animals to a sterilizing treatment previous to transplantation, and the whole transplantation process has been performed in autoclaved medium and under sterile conditions. Therefore, the bacteria found in the recipient animals at the end of the experiment should derive from the donors. To illustrate that convincingly on individual OTU level, we added Table S8 that shows the single bacterial OTUs that are shared between each donor and its recipient. It can clearly be seen that, while many bacterial species are shared among the most of the samples, some are unique for each specific AT and therefore, specifically transplanted.*

Line 505: change “to next generation” to “to the next generation”

Answer: done.

Line 505 (and 544): header suggests that microbiome and heat tolerance are transferred to the next generation but no comparison of donor recipient microbiomes is shown, and, heat tolerance was not assessed as for the previous microbiome transfer experiment. Wording to strong and conclusion is not justified by the data provided.

Answer: *We agree with the reviewer that we did not measure heat tolerance in the offspring generation because we did not conduct a heat stress experiment. Instead, we measured general offspring fitness by monitoring offspring survival at low, medium and high temperature. We changed the wording in the heading and the paragraph accordingly.*

In order to provide further evidences for the transmission, we added a PCoA in Figure S3 including both the F0 and F1 samples. We included also Table S11 that shows bacterial OTUs that are shared between each mother polyp and its offspring. It can be seen that, while many bacterial species are shared among the most of the samples, some are unique for each specific AT and therefore, specifically vertically transmitted.

We changed the corresponding paragraph in the following way (line 553-573): “In a second step, the F0 and the juvenile F1 polyps were subjected to 16S rRNA sequencing to evaluate the transmission of acclimated microbes to the next generation. PCoA of F1 animals revealed a significant clustering according to both F1 DT and F0 AT (Figure 6-c and d, Table 3). While, on average, around 50 % of bacterial variation can be explained by the DT of the juvenile polyps, around 20 % of the bacterial colonization in juveniles can be explained by the acclimation temperature of the F0 polyps (Table 3). The comparison of F0 and F1 samples in PCoA revealed a clustering between F0 and F1 samples along PC1 (Figure S3-a), indicating that differential bacterial communities colonize juvenile and adult polyps, as described in earlier studies 18, 19. In contrast, PC2 separated samples according to AT in both the F0 and the F1 samples (Figure S3-a, Table S10), indicating the successful transplantation of parts of the acclimated bacterial communities. The successful transmission is also indicated by the concordance of bacterial groups present in the F0 and F1 animals (Figure S3-b). For the identification of individual OTUs potentially transmitted from F0 to F1 animals, we performed a presence/absence analysis (Table S11) revealing the differential transmission of bacterial species from F0 female polyps acclimated at different temperatures.”

Line 544: the authors mention that “the fact that also parts of the acclimated microbiota is transmitted and persisting in the juvenile F1 polyps...”; however, they never show the microbiome composition of the juvenile F1 polyps. These data are required and could be added as an extra panel in Figure 6. Moreover, a statistical analysis of the differences in bacterial OTUs abundances, such as the one presented in Table S6, is required.

Answer: *We provided the requested data in a new panel in Figure S3-b. We added bar charts representing the proportion of bacterial groups in F0 and F1 samples. For statistical support, we performed an additional LEfSe analysis (new Table S12) supporting the conclusion that bacteria transmitted from the mother polyps persisted in the corresponding F1 polyps. We added these information in line 573-578.*

Line 544: Was the size of the F1 polyps also measured? I would assume AT 25°C derived polyps to be larger but is good information to have.

Answer: *No sorry, we did not measure the size of the F1 polyps, but no evident differences were detected.*

Line 566: There was bleaching, just lower than not acclimated corals.

Answer: corrected.

Line 572: This study does not disentangle the contribution of host gene expression and microbiota to temperature acclimation in cnidarians, as only one time point was collected for RNASeq data and it wasn't even under heat stress. It does, however, provide gene expression information of *N. vectensis* polyps acclimated to different temperatures at 75 woa and with different microbiota. It also provides insights into the effects of microbiome transplantation on heat tolerance in *N. vectensis*.

Answer: *We agree with the reviewer that we have not separated the two factors. We only separated the effects of bacterial plasticity from host effects by using microbial transplantation on a single genotype. We agree that with this experimental setup we were not able to estimate the contribution of host gene expression to temperature acclimation.*

We rephrased the corresponding sentence to: "Our study separates the contribution of the microbiota to temperature acclimation from host effects by performing microbial transplantation experiments in a single host genotype."

*In the discussion we added the following paragraph (line 688-695) : "Acclimation is generally thought to be driven by shifts in gene expression^{2,3}. Microbial transplantation experiments allowed us to measure the contribution of bacterial plasticity to host acclimation separately from genetic factors. Although we also observed significant adjustments in the transcriptomic response to thermal acclimation in *Nematostella*, we were unable to assess the contribution of changes in gene expression to thermal acclimation. The adjustments in gene expression are most likely a combination of acclimation to the new temperature condition and to the changed microbial colonisation."*

Line 594: Similar to the previous comment. This isn't disentangling gene expression and microbiota contribution, as gene expression is likely going to be different with different microbiota. But does remove the genotype effect by working with clones. More importantly, it does show the big impact of microbiota on survival.

Answer: *We agree again and changed the sentence to: "With this experimental setup, we were able to disentangle microbiota contribution to thermal acclimation from host genotype effects and proved that acclimated bacteria can act as heat tolerance promoting bacteria."*

Line 620: The paper that is cited does mention that there is an increased microbiota β -diversity in corals upon heat stress but argues the opposite to the point brought by the authors. Where increased β -diversity was associated with coral mortality, not tolerance. Additionally, the citation is given to a paper that cites another, the proper citation should be to: Zaneveld, J. R. et al. Overfishing and nutrient pollution interact with temperature to disrupt coral reefs down to microbial scales. *Nat. Commun.* 7, 11833 (2016).

Answer: *the sentence has been edited and the citation removed.*

Line 674: "and antifouling activity reference and reference found ..." Please check this sentence.

Answer: *corrected.*

Line 709: The results show an increase in survival at every temperature in offspring derived from the acclimated 25°C females (Fig. 6b). This can be observed at 15, 20, and 25°C. This suggests for an overall increase in survival of the offspring derived from 25°C acclimated mothers and not a thermal specific tolerance.

Answer: *The reviewer is right and we changed the sentence in the following way (line 748): "These results confirmed that polyps acclimated to high temperature, transmit a higher overall fitness to their offspring."*

Reviewer #2 (Remarks to the Author):

The manuscript entitled "Microbiota mediated plasticity promotes thermal adaptation in *Nematostella vectensis*" represents an impressively comprehensive long-term project studying the effects of temperature acclimation on a cnidarian holobiont. I found the experimental design appropriate for the

study question and the data analysis suitable for the claims. My suggestions for minor revisions for clarification are detailed below.

Methods:

Line 144: Clarify “smashing”. Is this homogenization? If so- what homogenization procedure was used? (e.g., mortar and pestle; bead beater; syringe; Dounce homogenizer; electric tissue grinder)

Answer: *yes, we homogenized the animals with the electric tissue grinder. We specified this in the M&M section.*

Line 154: Clarify what is the supernatant of acclimated adult polyps. The supernatant is from the anemone “smashing”? Please provide more detail here about the homogenization procedure and how the supernatant was obtained (centrifugation?).

Answer: *We specified the procedure in the M&M section.*

Lines 171–173; 179–180; and 192–193: Are the spawning animals the same animals that just came out of the heat stress or did these animals never enter the 40°C heat stress? Please clarify.

Answer: *The F0 animals used for spawning and offspring survival assay never entered the 40°C heat stress. For all these experiments acclimated clones were used, that were not stressed before. We clarified that in the corresponding sections (line 112).*

Line 195: Please clarify what “Five not induced female polyps” means in this sentence. Five females that did not enter the spawning cycle?

Answer: *we removed that part, as these samples were not included in the final analyses.*

Line 198: The abbreviation “DT” is used here and it seems like it could be have been introduced in line 176 for the other developing temperature earlier.

Answer: *We introduced the term “developmental temperature (DT)” in the section mentioned above.*

Figures:

Figure 1: This figure is greatly appreciated given the impressive long-term experimental design! I would find it helpful if there were more explanations for the sample size at each step. For example, “HS (1)” could be “HS 1 (n = x)”.

Answer: *We added the information regarding the replicates in the description of Figure 1.*

Figure 3: I might recommend combining panels A and B by using different shapes for 40, 70, and 132 weeks while keeping the colors for the different ATs. Either way the point is well illustrated but I did have trouble seeing the details of the figures because there is so much going on in Figure 3 (a good problem!).

Answer: *We agree with the reviewer on this point, that the panels are quite complex. However, to maintain colour consistency within the figure, we would like to keep the a and b panels separate. Otherwise, we would miss the counterpart in the colour scheme in panel e and g.*

Figure 5: Missing a closing parenthesis in the figure legend.

Answer: *done.*

Discussion:

Line 600: The authors make a good point about the animals being the same age and size. You observed considerable variation in size and cloning rate between the different acclimation temperatures. If you made an effort to track single polyps throughout time to make sure they were all the same age, please

note that here. Otherwise it seems that some polyps of the original 150 must have been older than others given that they originated from a single individual.

Answer: *The culture of animals used as recipients is a culture of animals of the same clonal line used for the acclimation experiment, but was maintained separated from the acclimated cultures under standard conditions. We did not track individual polyps in this culture. However, as it is a clonal culture without sexual reproduction, we can state, that all polyps have the same age. In addition, for transplantation we selected animals with similar sizes by eye.*

Minor edits:

Line 32: Typo: "... through genetic recombination and natural selection alo." Should be alone?

Answer: *done.*

Line 86: Typo: "Recently, has been shown that..." Missing a word?

Answer: *We added "it" here*

Line 109: Typo. No period between sentences ending in "USA" and starting with "They"

Answer: *done.*

Lines 135–139: This sentence is difficult to follow. I think there are unnecessary words that are making it difficult to understand (e.g., "and remove the most of associated bacteria").

Answer: *As the reviewer suggested, we deleted "and remove the most of associated bacteria" from the sentence.*

Line 200 and 207 and 217: Is "N" short for nitrogen? Saying nitrogen may be more clear or the chemical abbreviation LN₂.

Answer: *We replaced "N" with "nitrogen".*

Line 216: Typo: "Adult animals starved for 3 days..." missing the word "were"?

Answer: *done.*

Line 332: Typo: "...where performed at 40 and 132 weeks..." should be "were performed"

Answer: *done.*

Line 589: subscript 2 in carbon dioxide chemical abbreviation

Answer: *done.*

Line 595: "microbotas" should be "microbiota" or "microbes"

Answer: *corrected.*

Reviewer #3 (Remarks to the Author):

In this study the authors examine the effects of temperature acclimation, and the associated effects of the microbiome, on the survival of *N. vectensis* at higher temperatures. They also examine the transplantation of the microbiota and examine those effects. This study was well thought out, with appropriate controls and well-made figures. I especially appreciated the diagrams explaining the experimental procedures in Figures 1 and 6. The study incorporates not only 16S data, but also transcriptome data. It was a pleasure to read. The long-term duration of this experiment was also successful in showing how long it took for the microbiome to become stable in the different temperature conditions. The authors clearly demonstrate that the microbiome of *N. vectensis* acclimated at higher temperatures long-term are better equipped to cope with temperature stress. Work linking cnidarian adaptation to microbiome to changing temperatures is much needed, and long-term experiments of this

sort are rare. My comments are all fairly minor, and mostly have to do with clarity. I think this is a really well done paper, and strongly support publication. Great job!

Comments:

1. In line 56 you refer to the Modern Synthesis. While this is a concept that most should be familiar with, my quick poll of some colleagues (especially those on the younger side) suggests that maybe it is less well known (at least by this term) than it should be. I'd suggest citing the original work.

Answer: *We added the original citation*

2) You are using a lot of acronyms in this manuscript, and maybe don't need all of them. I would probably try to spell out woa and AT in each figure legend, and other acronyms like HTPB are only used a couple of times and could just be spelled out to avoid confusion. Same with ME, BMC, AE, MMTA.

Answer: *We reduced the use of acronyms. The remaining ones we introduced in the figure legends.*

3) In Figure 2 I think it was a little unclear if the sample in B/C/D were being held at the AT throughout – I might include that info in the legend. (I was able to figure it out from the text, but I think it's good for the figures to be able to stand on their own.)

Answer: *We included the information in the legend.*

4) In Figure 3, I both like and dislike that the PCoA was shown twice with different grouping. I think you're making an important point and having them shown twice is useful for that, but I think it could be worthwhile to indicate in some way that the two figures show the same samples. At a minimum I'd state this in the legend, but it might be nice to indicate woa with shape in A, and AT with shape in B, or something like that. I think the same comment applies for Figure 6 C and D. (I liked that you were consistent with color choice throughout, so not suggesting you change that!)

Answer: *We agree with the reviewer that the dual presentation of data could be confusing. Therefore, we have included the information as a note in the figure legend. We have tried to include different shapes in these figures, but we have concluded that this overloads the information in the charts.*

5) The one result I found surprising was the low survival of polyps acclimated to 25°C after heat stress, which resulted in many more dead polyps after 40 weeks, although not at 132 weeks. You mention a possible explanation for this in lines 560-561 (although I think you should clarify in those lines that you are discussing the 40°C heat stress). I was thinking that the transplantation experiments would support this experiment as well, but I don't think you did the strong heat shock on those samples. Looking more closely at Figure 3, I'm still struggling with this a bit, because at the 40 woa, the microbiome of the 25 and 20 AT still cluster together fairly closely, so it doesn't seem like the microbiome is distinct enough to explain such a strong change in survival rates. I'd appreciate more discussion of this point in the discussion.

Answer: *We agree with the reviewer and struggled with these results as well. The reviewer is correct in that the transplantation experiment performed with the donor microbiota of 132 woa supports the second heat stress experiment that was also performed after 132 woa. In both cases, the heat stress conditions were the same, and the animals of 25 °C had the highest survival rate. We also included the information of the heat stress information in the legend of Figure 5.*

Another explanation for the discrepancy between survival rates and differences in microbial community composition could be that we only detected changes in bacterial abundances with 16S rRNA gene sequencing. It is likely that not only changes in bacterial abundance, but also changes in bacterial gene content affect stress resistance. However, these changes cannot be traced based on 16S data, but only by metagenomic analyses. We already discussed that point in the discussion: line 655-662.

Minor comments:

Line 32: ...natural selection also...

Answer: done.

Line 34: ... microbiota as a putative source...

Answer: done..

Line 56: I felt that

Answer: *It is not clear what the reviewer meant by this comment.*

Line 76: burrowed, rather than borrowed

Answer: done.

Line 121: maybe “the polyp number was maintained consistently”

Answer: *Here, we mean “constant”.*

Line 123: were, rather than where

Answer: done.

Line 320: acclimated, rather than acimated

Answer: done.

Line 328: temperatures, rather than temperature

Answer: done.

Line 347: were, rather than was

Answer: done.

Line 357: I think the temperature here should be 40°, not 25° □ “the mortality rate at 25°C was significantly reduced”

Answer: *Here, we are not writing about the heat stress but the survival along the acclimation process of 161 weeks. To make that more clear, we added this information to the legend and to the text.*

Line 348: I think it should say “per box” rather than “and box”

Answer: done.

Line 348: Refers to panel E which isn't present – I think it should be C

Answer: done.

Line 381: binary, rather than pinary

Answer: done.

Line 437-438: ...showing the number of differentially expressed genes within the three AT...

Answer: done.

Figure 6: The AT designation in B looks like it belongs to 20°C only. It might be clearer to put AT above those three temperatures.

Answer: done.

Line 573: cnidarians, rather than cnidarian

Answer: done.

Line 597: before, rather than until

Answer: We changed the sentence in the following way: “Microbial community acclimation is a highly dynamic process that began in the first few weeks after environmental change, and most adjustments in bacterial diversity occurred by 84 woa.”

Line 609: I found the phrase “punctual abundances” to be hard to interpret here – I would revise to clarify your meaning.

Answer: We changed the sentence into: “..., while a more dynamic part of the microbiota changed by either increasing or decreasing abundance of certain species.”

Line 611: ...from the surrounding environment, or a higher...

Answer: we changed to “environment” only.

Line 734-736: I'd revise this sentence to maybe something like: “Recently, corals subjected to experimental warming and inoculated with consortia of potentially beneficial bacteria were shown to bleach less when compared to corals that received no probiotics.” (otherwise need to change the beginning as you are missing a noun)

Answer: We changed the sentence to: “Recently, corals exposed to experimental warming and inoculated with consortia of potentially beneficial bacteria were shown to bleach less when compared to corals that did not receive probiotics.”

Line 740: I would change the semicolon to a comma, and add an additional comma later in the sentence: ...heat stress, and that this resistance...

Answer: done.

References cited

Reitzel, A. M., Chu, T., Edquist, S., Genovese, C., Church, C., Tarrant, A. M., & Finnerty, J. R. (2013). Physiological and developmental responses to temperature by the sea anemone *Nematostella vectensis*. *Marine Ecology Progress Series*, 484(Somero 2012), 115–130. <https://doi.org/10.3354/meps10281>

REVIEWERS' COMMENTS

Reviewer #1 (Remarks to the Author):

The authors have addressed most of my comments adequately and the revised manuscript is significantly improved. I only have a few minor comments regarding the wording in some statements.

Line 45: There was no indication of transmission of thermal resistance to the next generation as this was never tested and offspring from 25 °C acclimated parents showed the same increased survival irrespective of the temperature they developed in. Therefore, what was observed is basically an overall increase in fitness of offspring derived from 25 °C acclimated individuals.

Line 344 and Figure 2-a: The significance values drastically changed in this panel relative to the previous version of the manuscript, yet the test is mentioned to be the same. Why the discrepancy? While the new values make more sense for 132 woa (as before the values were barely significant between 100% survival and 0%), they now show no difference between 20 and 25 °C acclimated polyps at 40 woa. If this is the case, it might be appropriate write in line 344 after “of 70 % and 30 %, respectively”, although not significantly different.

Tables S8 and S11 in the excel file have their headers naming them as S7 and S9

Tables S8 and S11: Are very informative tables. Would it possible to separate 0 into two numbers to distinguish those OTUs that were only found in the donor and those only found in the recipient?

Lines 742-743 (and line 752): As mentioned above, heat tolerance was never really tested in the offspring, so it is not possible to conclude that thermal resistance was passed on to them. What has been shown is an increase in overall fitness at “higher” temperature that was passed on to the offspring, either through the associated bacteria, epigenetic modifications, or a combination of both.

Reviewer #3 (Remarks to the Author):

I thank the authors for their edits, and think the manuscript is now ready to be published.

Reviewer #1 (Remarks to the Author):

The authors have addressed most of my comments adequately and the revised manuscript is significantly improved. I only have a few minor comments regarding the wording in some statements.

Answer: we thank the reviewer for appreciating our work that we think benefitted from his/her suggestions. In the following we address the further comments.

Line 45: There was no indication of transmission of thermal resistance to the next generation as this was never tested and offspring from 25 °C acclimated parents showed the same increased survival irrespective of the temperature they developed in. Therefore, what was observed is basically an overall increase in fitness of offspring derived from 25 °C acclimated individuals.

Answer: We have rephrased accordingly and shortened the abstract to less than 200 words and converted to present tense as required by Nat. Comms. guidelines.

Line 344 and Figure 2-a: The significance values drastically changed in this panel relative to the previous version of the manuscript, yet the test is mentioned to be the same. Why the discrepancy? While the new values make more sense for 132 woa (as before the values were barely significant between 100% survival and 0%), they now show no difference between 20 and 25 °C acclimated polyps at 40 woa. If this is the case, it might be appropriate write in line 344 after “of 70 % and 30 %, respectively”, although not significantly different.

Answer: The reviewer is right, that the significance bars changed between the different version of the manuscript. During revision we realized, that the stars shown did not fit to the statistical values of the test performed. After reevaluating we now presenting the stars fitting to the p values. As requested, we now present the details to the statistics including the p-values into the figure legend. As requested by the reviewer, we added the sentence in the manuscript accordingly. We apologize for these mistakes.

Tables S8 and S11 in the excel file have their headers naming them as S7 and S9

Answer: we corrected accordingly

Tables S8 and S11: Are very informative tables. Would it possible to separate 0 into two numbers to distinguish those OTUs that were only found in the donor and those only found in the recipient?

Answer: We thank the reviewer for this suggestion and we added this additional information to both tables. We now indicated if an OTU is present in both sample categories or only in one. If the OTU is present only in one sample, we indicated in addition in which of the both categories.

Lines 742-743 (and line 752): As mentioned above, heat tolerance was never really tested in the offspring, so it is not possible to conclude that thermal resistance was passed on to them. What has been shown is an increase in overall fitness at “higher” temperature that was passed on to the offspring, either through the associated bacteria, epigenetic modifications, or a combination of both.

Answer: We adjusted both sentences as requested.